

# Energy efficiency in transient surface runoff and sediment fluxes on hillslopes – a concept to quantify the effectiveness of extreme events

Samuel Schroers[1], Ulrike Scherer[2], Erwin Zehe[1]

[1]Institute of Water Resources and River Basin Management, Karlsruhe Institute of Technology – KIT, Karlsruhe, Germany
[2]Engler-Bunte-Institut, Water Chemistry and Water Technology – KIT, Karlsruhe, Germany

*Correspondence to*: S. Schroers (samuel.schroers@kit.edu)

**Abstract.** Surface runoff over time shapes the morphology of the landscape. The resulting forms and patterns have been shown to follow distinct rules, which hold throughout almost all terrestrial catchments. Given the complexity and variety of the earth's runoff processes, those findings have inspired researchers for over a century, and they resulted in many principles and sometimes proclaimed laws to explain the physics that govern the evolution of landforms and river networks. Most of those point to the $1^{st}$ and $2^{nd}$ law of thermodynamics, which describe conservation and dissipation of free energy through fluxes depleting their driving gradients. Here we start with both laws but expand the related principles to explain the coevolution of surface runoff and hillslope morphology by using measurable hydraulic and hydrological variables. We argue that a release of the frequent assumption of steady states is key, as the maximum work that surface runoff can perform on the sediments relates not only to the surface structure but also to "refueling" of the system with potential energy by rainfall events. To account for both factors, we introduce the concept of relative dissipation, relating frictional energy dissipation to the energy influx, which essentially characterises energy efficiency of the hillslope when treated as an open, dissipative power engine. Generally, we find that such a hillslope engine is energetically rather inefficient, although the well-known Carnot limit does not apply here, as surface runoff is not driven by temperature differences. Given the transient and intermittent behaviour of rainfall runoff, we explore the transient free energy balance with respect to energy efficiency, comparing typical hillslope forms that represent a sequence of morphological stages and dominant erosion processes. In a first part, we simulate three rainfall-runoff scenarios by numerically solving the shallow water equations and we analyse those in terms of relative dissipation. The results suggest that older hillslope forms, where advective soil wash erosion dominates, are less efficient than younger forms which relate to diffusive erosion regimes. In the second part of this study, we use the concept of relative dissipation to analyse two observed rainfall runoff extremes in the small rural Weiherbach catchment. Both flood events are extreme, with estimated return periods of 10000 years and produced considerable erosion. Using a previously calibrated, distributed physics-based model, we analyse the free energy balance of surface runoff simulated for the 169 model hillslopes and determine the work that was performed on the eroded sediments. This reveals, that relative dissipation is largest on hillslope forms which relate to diffusive soil creep erosion, and lowest for hillslope profiles relating to advective soil wash erosion. We also find that power in surface runoff and power in the complementary infiltration flux are during both events almost identical. Moreover, there is a clear hierarchy of work, which surface runoff expended on the sediments and relative dissipation between characteristic hillslope clusters. For



hillslope forms that are more energy efficient in producing surface-runoff, on average a larger share of the free energy of surface runoff performs work on the sediments (detachment and transport) and vice versa. We thus conclude that the energy efficiency of overland flow during events does indeed constrain erosional work and the degree of freedom for morphological

changes. We conjecture that hillslope forms and overland dynamics coevolve, triggered by an overshoot in power during intermittent rainfall runoff events, towards a decreasing energy efficiency in overland flow. This implies a faster depletion of energy gradients during events, and a stepwise downregulation of the available power to trigger further morphological development.

## 1 Introduction

Water-rock interactions, chemical weathering and fluvial erosion have relentlessly shaped our Earth over the past 3.8 billion years (Wolman and Miller, 1960). By performing physically work on the land surface, overland flow erodes and transports sediments, thereby shaping landforms and fluvial networks with distinct characteristics at almost any scale. Prominent examples thereof are expressed in Horton's laws of stream number, area and length (Shreve, 1966) or Hack's law about the upslope contributing catchment area and maximum stream length (Hack, 1957). Moreover, there is a distinct relation between

the size and return period of flood peaks and the channel cross section (Leopold and Maddock, 1953), as well as shape and extend of the flood plain and sediment transport (Dunne et al., 1998). At the hillslope scale, one can depending on the morphological age of the system observe typical hillslope forms. These reflect the dominant erosion processes of diffusive soil creep, rain splash and advective soil wash (Kirkby, 1971 or Bonetti 2020). Thus, on the catchment as well as hillslope scale, surface runoff dynamics and geomorphic features are co-organized in a highly complex manner. Due to the complexity of

these interactions and their multiple scale dependent manifestations many concepts to explain the co-evolution of surface runoff and landscape morphology are of semi-empirical nature. This implies that they partly rely on "tuning" parameters, which capture the relation between fluid flow and channel or hillslope geometry, as well as physical properties for a particular environmental and hydro-climatological setting and scale (Wolman and Gerson, 1978; Beven, 1981). However, despite of these obstacles, there has been continuous research to discover the seemingly hidden physical laws governing and constraining

the co-development of form and functioning of the Earth's hydrologic systems (Leopold and Langbein, 1962; Yang, 1971; Riggs, 1976; Wolman and Gerson, 1978; Dietrich et al., 1982; Howard, 1990; Rodriguez-Iturbe et al., 1992; Perron et al., 2009).

In line with the idea that morphological changes of the land surface require physical work (Wollman and Miller, 1960), these

studies relate observed spatial patterns to the directed evolution of the system (river network, catchment or hillslope) towards a steady state optimum configuration. For these cases, optimality refers in some sense to the dissipation of free energy in an open system, leading in the context of a stream to the local maximization of stream power (Kleidon et al., 2013) and to the minimum (free) energy expenditure of average discharge in the stream network as a whole (Rodriguez-Iturbe et al., 1992). On





the hillslope scale Zehe et al. (2010 and 2013) showed that macropore flow patterns relate to maximum free energy dissipation

and correspond to maximum entropy production (Leopold and Langbein, 1962). The fundamental reason why free energy can be dissipated and hence be lost to the process dynamics, arises from the 2$^{nd}$ law of thermodynamics. The latter states that entropy cannot be consumed, but it is produced during irreversible processes. At a very basic level this implies that fluxes deplete their driving gradients (and that water flows downslope). Although energy is conserved and cannot disappear due to the 1$^{st}$ law of thermodynamics, free energy is not a conserved property, but is dissipated during irreversible processes due to

the related production of entropy. Free energy is basically energy without entropy, and the free energy of a flow system is thus equivalent to its capacity to perform work to steepen a concentration gradient (Zehe et al, 2021) or to create motion in form of coupled water and sediment fluxes (Bagnold, 1966). Frictional dissipation during the latter implies production of heat through production of entropy, which increases the average kinetic energy of the molecules in the riverbed or the hillslope surface materials. As heat corresponds to a random isotropic motion of molecules it cannot be converted (back) into work to generate

overland flow by cooling down the riverbed. While this would be consistent with energy conservation, it would violate the second law as it required consumption of entropy. Any increase in entropy of an isolated environmental system goes hence on the expense of a reduction of available free energy and the system's capacity to perform work. This implies that the system ends in a dead state called thermodynamic equilibrium where all gradients have been depleted, corresponding to minimum free energy and maximum entropy. Open thermodynamic systems may however prevail in an organized state far away from

the entropy maximum, if there is an external feedback sustaining a net influx of free energy to perform the necessary work to act against the depletion of gradients and to export the entropy produced during irreversible processes (Zehe et al., 2021). In the following we want to clarify this aspect for surface runoff and related ideas of thermodynamic optimality, which appear to be contradictory at first sight.

The potential energy of water molecules and the related flux of potential energy is clearly larger at the upstream/upslope end of its flow path where the atmosphere re-delivers water via rainfall to the land, than at its downstream/ downslope outlet where water runs off to the sea/ or the river. This free energy difference is characterized by the geopotential gradient along the hillslope/ river course on one hand and the downstream/downslope accumulation of runoff/water mass on the other hand (Schroers et al., 2022). Both factors jointly determine the maximum amount of work the system could perform in a mechanical

sense (Gillet, 2006). We thus speak of the free energy of surface runoff. However, as pointed out by previous studies (Schroers et al., 2022; Loritz et al. 2019) only a minute amount of this free energy is actually converted into work i.e., the kinetic energy of the coupled water and sediment flux, while the vast majority has dissipated at the downstream/downslope outlet. Recalling the concept of energy efficiency, which relates the work per time i.e., the power produced by a heat engine/ power plant to the energy influx, surface runoff has a very low energy efficiency, at least during steady state flow conditions. This is striking,

because the energy efficiency of surface runoff is not limited by the well-known Carnot limit. The latter is generally valid for heat engines, and it also limits turbulent fluxes of sensible heat in the atmosphere (Kleidon et al., 2018; Conte et al., 2019).




Runoff is however not driven by a temperature gradient but a gradient in geopotential. Rainfall and tectonic uplift distribute water and sediments against the geopotential gradient, thereby maintaining a permanent disequilibrium in the coupled water and sediment cycles in river basins by "refueling the catchment engine". These open systems can hence evolve towards an

optimal configuration far from the entropy maximum (Kleidon, 2016): the periodic and intermittent input of free energy by rainfall results in co-adaptive development of the internal structure and the space-time pattern of water and sediment fluxes. In this context Leopold and Langbein (1962) put the river in analogy to a chain of heat engines and showed that maximization of entropy production by stream flow must result in an exponential geo-potential profile of a rivers' course through the landscape, which can indeed be found for many rivers (Langbein, 1964; Tanner, 1971). While this study is certainly a landmark

and the analogy is appealing, the reasoning is not fully consistent, as runoff is not driven by temperature gradients and the Carnot limit does not constrain energy efficiency. Later onwards, Yang (1971) introduced the minimum stream power theory, which was placed on the minimum entropy production concept proposed by Prigogine (1946) in physical chemistry. Rodriguez-Iturbe et al. (1992) extended this work to the theory of optimal channel networks by postulating three principles: (1) the principle of minimum energy expenditure in any link of the network, (2) the principle of equal energy expenditure per

unit area, and (3) the principle of minimum total energy expenditure in the entire river network. These principles apply to steady state and thus average discharge conditions, which assures that the constraints of a closed catchment water balance is fulfilled. The inconsistency here is that bank full discharge corresponds according to Wollman and Miller (1960) to the two to ten years flood and not to average discharge. If the channel is formed by fluvial erosion, this implies that the kinetic energy balance of the sediments is not included in this theory, as average discharge is less than bank full discharge. In a later work

Wolman and Gerson (1978) extended the idea to effective landscape forming events and added the notion that dynamic thresholds determine the effectiveness of a runoff event, leading to event sequencing (Beven, 1981).

More recently, Kleidon et al. (2013) applied the maximum power principle, originally proposed by Lotka (1922), to river systems and proposed that those develop to a state of maximum power in the coupled water and sediment flux. They argued, while the driving geopotential gradient is depleted at the maximum rate, the associated sediment export maximizes with the

same rate. The weakness of this analysis was to treat the catchment as runon-runoff system, where water is added at the uppermost stream segment as a constant discharge along the course of a river. Catchments are however mass accumulative because they receive their rainfall in a spatially distributed manner, resulting in downstream growth of stream flow (Schroers et al., 2022). This means that in the upper part of the slope/catchment potential energy of surface runoff grows in downslope direction to a local maximum and declines afterwards. Moreover, maximum power in the combined sediment-water flux does

in steady state correspond to maximum entropy production. This idea hence seems to contradict the idea of minimum energy expenditure assuming minimum entropy production.

These apparent contradictions can be explained by at least two pitfalls that emerge, when working with the analogy to heat flows and entropy production in geosciences. First, there exist at least three forms of physical entropy (not to mention

information entropy), (cf. Popovic, 2017) namely thermal entropy produced by depletion of temperature gradients, molar



entropy produced by mixing and depletion of chemical potential and geo-potential gradients, and radiation entropy produced by radiative cooling (Kleidon, 2016, Zehe et al., 2021). And second, a proper definition of entropy production requires a clear definition of the system and its boundary, otherwise "Nobody really knows what entropy is" (Von Neumann, cited in Tribus and McIrving, 1971). In this light, minimum energy expenditure refers to the production of thermal entropy through friction,

which shall be minimized in the entire network. Minimum dissipation results in maximum power of stream flow, as energy is conserved. This implies in turn a maximum flux of water (and sediments) and thus maximum production of molar entropy. We therefore very much agree with e.g. Kleidon (2016) that an exact definition of the system and a proper terminology which kind of entropy is produced in which part of the system, resolves these apparent contractions.

In line with these thoughts, we propose here that the concepts of free energy, work and energy efficiency are much more suited for analyzing the interplay of (land-) form (-s) and functioning of overland flow systems. Starting point is our previous work (Schroers et al., 2022), which revealed that the aforementioned morphological stages and related typical hillslope forms, do not only reflect the transition of the dominant erosion processes from diffusive soil creep, over mixed behavior to advection dominated soil wash (Kirkby, 1971), but are also a manifestation of a hierarchy of energy efficiency of overland flow. This

can be explained by the fact that a change of the longitudinal hillslope profile affects not only the driving geo-potential gradient, but also the amount of rainfall that is locally intercepted by the projected area on the horizontal axis. We defined relative dissipation as dissipated fraction of free energy of overland flow, normalized by energy influx due to precipitation. Relative dissipation was largest for hillslope profiles relating to soil wash erosion and minimum for profiles where soil creep dominates. This suggests that hillslope forms develop towards smaller energy efficiency in overland flow, meaning that a larger fraction

of the energy influx is dissipated for hillslopes which are closer to a dynamic equilibrium than for hillslopes which are far away from an equilibrium. In consideration of the effectiveness concept of hydrological events, coined by Wolman and Gerson (1978), relative dissipation also captures the notion of dynamical thresholds (cf. Beven, 1981) and beyond that gives us a useful starting point for a thermodynamic evaluation of these.

We furthermore showed that the emergence of rills increases the power and thus the energy efficiency in steady state overland

flow, but also relates to larger friction coefficients which in turn limit overall energy efficiency. This feedback resulted in maximum relative dissipation or equivalently minimum relative free energy at the outlet and showed a correlation with sediment transport rates. Here we step beyond this analysis by releasing the steady state assumption, which is rarely fulfilled during natural rainfall events. This is particularly true for hillslopes because overland flow events are intermittent. It is important to extend our concept to transient conditions, because in steady state dissipation in overland flow is almost equal to

the power input. Structural development needs however work and thus an overshoot in power, meaning a certain resistance threshold must be exceeded for effective erosion events (Wolman and Miller, 1960).

Steady state hydraulic conditions imply time invariant flow depths (Chow, 1959). This is seldomly achieved in natural streams and practically non-existent for overland flow and surface runoff on hillslopes (cf. Dunne and Dietrich, 1980; Emmett, 1969).



Yet, most laboratory (Gimenez and Govers, 2002; Rieke-Zapp and Nearing 2005) and field experiments (Nearing et al., 1997) studying surface runoff on hillslopes have been set up in a way to reach steady state conditions and conclusions are drawn from adaptations to this state. Time is even more important, when considering the interaction of the water fluid with sediments. For rivers it is well known that sediment transport is directly coupled to unsteady state flood waves, which trigger the detachment of larger particles leaving smaller particles unprotected and prone for transport (Gob et al., 2010). Similar

behaviour was shown by Kinnell (2020) for hillslopes, where the onsets of particle detachment and transport are distinctly linked to different points in time during surface runoff events. Importantly, steady states considering coupled fluid and sediment fluxes differ considerably from those dealing only with fluid flow. This is firstly due to the transport mechanism which governs sediment travel times and can lead to much slower sediment particle velocities than water flow. And secondly, transient loads of suspended particles imply a changing fluid density, even if fluid and particle velocities would not change

with time. A true steady state is therefore not achieved until the slowest moving particle detached at the point farthest from the discharge point is discharged at the outlet and a continuous steady sediment transport is reached. This requires obviously periods of time invariant rainfall, otherwise transport and therefore time of concentration of sediment discharge might be altered.

This study has hence two objectives. First, we expand our thermodynamic framework for analysing the free energy balance of transient surface runoff and sediment flows using measurable hydraulic flow parameters. To this end, we simulate surface runoff events using the above mentioned characteristic 1D hillslope profiles, which relate to different dominant erosion and relative dissipation regimes. We use the 1D shallow water equations for this purpose, because they do not rely on a quasi-steady state momentum balance, and we apply a finite difference McCormack time diminishing variation (TVD) scheme to

numerically solve it. The benefit from this is a more accurate simulation of flow velocities and thus kinetic energy, which assures a more reliable calculation of the transient free energy balance of surface runoff, as well as the related energy efficiency to test our hypothesis about a power maximum in time. In a second part of the paper, we apply our theory to two rainfall runoff extremes observed in the Weiherbach catchment. To this end we employ an existing setup of the Catflow model (Zehe et al., 2005), representing the catchment by 169 typical hillslopes, accounting for the pattern of crops and their roughness, and an

interconnected river network. We use this simulated surface runoff for comparing relative dissipation and erosion patterns between characteristic hillslope types. Although the morphological development in the Weiberbach catchment has been affected by anthropogenic land use, we hypothesize that specific hillslope morphologies show distinct fingerprints of relative dissipation and power of transient surface runoff as well as sediment transport.



## 2 Theory

### 2.1 The hillslope as open thermodynamic system


The theory and applications of this paper are an extension to our first publication (Schroers et al., 2022) regarding steady state dissipation regimes. Therefore, we present here the final equations only and refer to our study for details. In general, we represent the hillslope surface as an open thermodynamic system (OTS) (Kleidon 2016, Zehe 2013), which exchanges mass, momentum, energy, and entropy with its environment. The boundaries of the system are a subjective choice, depending on the

type and objectives of the analysis and are defined here as the hillslope surface without its subsurface soil structure (compare e.g., Zehe et al., 2013), starting at the topographic divide upslope and ending at the drainage channel downslope. Within these boundaries, we set surface runoff into a thermodynamic perspective and apply the first and second law of thermodynamics, which constitute that energy is conserved and entropy of an isolated system can only grow (Kondepui and Prigogine, 1952). We start with the assumption that a hillslope can be defined as a spatially integrated OTS, here denoted by the subscript $HS$

(cf. Fig. 1).

Energy dynamics of this OTS black box are therefore driven by a single representative influx of potential energy $J_{HS,in}^{pe}(t)$ in watt, on hillslopes in the form of rainfall, which leads to spatial gradients of geopotential of water. Over a certain flow path distance $L_{HS}$, these gradients are then converted into kinetic energy $E_{HS}^{ke}(t)$ in Joule (surface runoff) and heat, which is composed of changes in temperature and entropy. These spatial dynamics in time lead to Eq. (1), with net potential energy

flow $J_{HS,net}^{pe}(t)$ (watt) (Eq. (3)), either increasing potential energy of the system $E_{HS}^{pe}(t)$ or powering the creation of another type of energy $P_{HS}(t)$ (watt). Eq. (2) details how $P_{HS}(t)$ either leads to the creation of kinetic energy or dissipation $D_{HS}(t)$. Additionally, net kinetic energy flow $J_{HS,net}^{ke}(t)$ accounts for the net gain or loss of kinetic energy flow of the system. Eq. 1 to 4 are a simplification of surface runoff, as we do not consider other types of energy than potential and kinetic energy of water. For the here presented applications however all other energy types can be considered negligible.

$$\frac{dE_{HS}^{pe}(t)}{dt} = J_{HS,net}^{pe}(t) - P_{HS}(t) \tag{1}$$

$$\frac{dE_{HS}^{ke}(t)}{dt} = P_{HS}(t) - D_{HS}(t) + J_{HS,net}^{ke}(t) \tag{2}$$

$$J_{HS,in}^{pe/ke}(t) - J_{HS,out}^{pe/ke}(t) = J_{HS,net}^{pe/ke}(t) \tag{3}$$

In combination Eq. (1) and (2) lead to Eq. (4), which relates in- and output of energy of a system with the energy stored within the system.

$$\frac{dE_{HS}^{pe}(t)}{dt} + \frac{dE_{HS}^{ke}(t)}{dt} = J_{HS,net}^{pe}(t) + J_{HS,net}^{ke}(t) - D_{HS}(t) \tag{4}$$



For a rainfall runoff event the black box OTS (Fig. 1, Eq. 4) of a hillslope surface can be further simplified. We assume that the system receives on its upper end a constant potential energy inflow $J_{HS,in}^{pe}(t)$ and releases a time dependent energy outflow

at the lower end. We assign the lower end a bed level of zero, which makes the specific geopotential of the lower boundary flux only dependent on the water depth. In this case we regard the potential energy which enters the system much larger than the potential energy which leaves the system and therefore also $J_{HS,out}^{pe}(t)$ (watt) to be negligible. The kinetic energy flow at the inflow boundary is also assumed to be zero and temporal gradients are abbreviated by dot notation (e.g., $\frac{dE_{HS}^{pe}}{dt} = \dot{E}_{HS}^{pe}$) so that we can write the reduced equation 4 as:

$$D_{HS}(t) = J_{HS,in}^{pe}(t) - \dot{E}_{HS}^{pe}(t) - J_{out}^{ke}(t) - \dot{E}_{HS}^{ke}(t) \tag{5}$$

Each of the terms of Eq. 5 shall be derived from integration of spatially distributed hydraulic flow variables. For a detailed calculation of spatially distributed steady state dynamics, we refer to Schroers et al. (2022), and for the derived transient system a summary is presented in Appendix A.

From Eq. 5 we deduce that a transient system has several degrees of freedom to in- or decrease dissipation rates (or free energy

respectively), whereas a steady state system can only adjust the outflux of kinetic energy ($J_{out}^{ke}$). For the transient case, the influx potential energy can also be converted into potential and kinetic energy, stored within the system itself. For a constant energy influx, power can e.g., be maximized through minimization of increases in $E_{HS}^{pe}$, meaning less influx energy is converted into potential energy and more into kinetic energy. It is therefore possible that a system maximizes power whilst also minimizing dissipation. It is tempting to think that this simplification holds for discrete timesteps, but as natural systems are

highly transient it seems more likely that total dissipation in time or a maximum value during a concrete time-interval might be optimized. If a system receives a certain amount of energy influx, it is therefore clear that optimization must happen through adjustment of the internal spatial structure which determines temporal derivatives of free energy conversion rates. Previously (Schroers et al., 2022) we assumed the system to be in steady state and analysed the local adjustment of free energy conversion rates. Dissipation can be minimized by geomorphological adaptations of the hillslope surface optimizing loss of energy per

wetted cross section. For a transient event we integrate over a spatial domain and have an additional degree of freedom as energy can be stored in time. We therefore expect that the structure of hillslopes is not a result of a steady state but rather an outcome of many transient events (cf. Wolman and Gerson, 1978), during which free energy gradients are depleted as fast as possible.



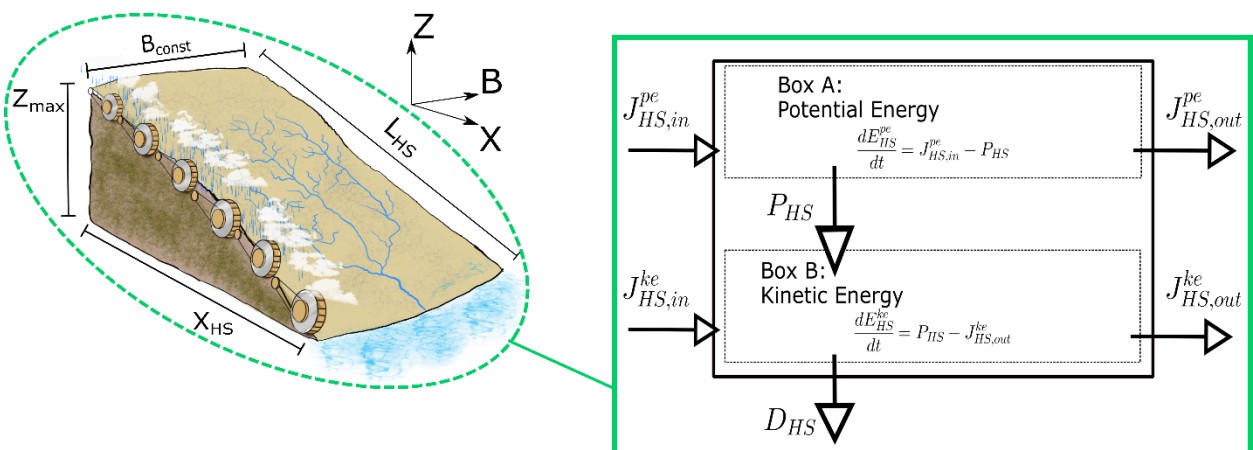

**Figure 1: Representation of energy conversion processes of surface runoff on hillslopes for a spatially integrated system**

## 2.2 Relative Dissipation of surface runoff

As hillslopes vary spatially in vertical as well as horizontal length scales and surface runoff events vary in time, absolute values do not represent relative dynamics of the energy balance and need to be normalized for comparison. Starting with Eq. 5, we first calculate the accumulated dissipation $D_{HS}^{acc}$ (joule, Eq. 6) for an event from $t=0$ to $tl$ and then normalize by the influx of energy $J_{HS,in}^{acc} = \int_{t=0}^{tl} J_{HS,in}^{pe}(t)\,dt$ which is accumulated at time $tl$ (Eq. 7).

$$D_{HS}^{acc} = \int_{t=0}^{tl} (J_{HS,in}^{pe} - \dot{E}_{HS}^{pe} - \dot{E}_{HS}^{ke} - J_{HS,out}^{ke})\,dt \tag{6}$$

$$\widehat{D}_{HS} = \frac{D_{HS}^{acc}}{J_{HS,in}^{acc}} = 1 - \frac{\int_{t=0}^{tl} (\dot{E}_{HS}^{pe} + \dot{E}_{HS}^{ke} + J_{HS,out}^{ke})\,dt}{J_{HS,in}^{acc}} \tag{7a}$$

$\widehat{D}_{HS}$ is dimensionless (joule joule$^{-1}$) and represents a thermodynamic descriptor for a spatially defined system which can be analysed in time for a given rainfall-runoff event. In the following we apply Eq. 7a for comparison of relative dissipation rates for characteristic hillslope profiles. The energy influx normalization is useful as it allows a comparison of different transient rainfall-runoff events independent of absolute rainfall rates and vertical as well as horizontal hillslope lengths. The second term on the right side of Eq. 7a can also be termed energy efficiency of overland flow, a larger value leads to less relative dissipation and reversely a lower value increases $\widehat{D}_{HS}$. Maximum relative dissipation is therefore related to minimum energy efficiency. Additionally we define relative stored energy $\widehat{E}_{HS} = \int_{t=0}^{tl} \frac{\dot{E}_{HS}^{pe} + \dot{E}_{HS}^{ke}}{J_{HS,in}^{pe}}\,dt$ as well as relative energy flux at the hillslope foot as $\hat{J}_{HS} = \int_{t=0}^{tl} \frac{J_{HS,out}^{ke}}{J_{HS,in}^{pe}}\,dt$, leading to a shortened version of Eq. 7a:

$$\widehat{D}_{HS} = 1 - \widehat{E}_{HS} - \hat{J}_{HS} \tag{7b}$$





## 3. Energy efficiency of transient overland flow as a function of hillslope form and erosion process

In this first part of the study, we test our hypothesis that the evolution of landscape forms is directly linked to energy efficiency of transient overland flow events. In its simplest form, the distribution of geopotential gradients can be related to prevalent erosion processes, ranging from very diffusive erosion regimes (soil creep, rain splash) to more advective flow regimes (soil wash, river flow) (Kirkby, 1971). These erosion regimes are per definition directly linked to the effectiveness of overland flow to erode and transport soil particles. Soil creep related hillslopes are therefore likely to have seen significant overland flow less

frequently and on smaller magnitudes, while the opposite can be said of hillslopes related to soil wash. This hierarchy should consequently translate into differences in energy conversion rates, resulting in some optimization with regard to overland flow on soil wash related profiles. To test this idea, we use the existing theory about erosion processes and hillslope form to construct characteristic 1D hillslopes and analyse overland flow scenarios on these within the context of energy efficiency. Transient overland flow is modelled by numerically solving the 1D Saint Venant equations through a McCormack scheme (Liang, 2006)

on a space time grid.

### 3.1 Erosion process and hillslope form

Quantitative geomorphological modelling is concerned with the development of landforms, given some initial and idealized boundary conditions (e.g. Wilgoose, 1991; Perron et al., 2009). Typically, the form of a hillslope is being modelled by solving partial differential equations of sediment and water mass conservation, coupled by semi-empirical transport laws (Beven,

1996). The parameters of these laws are usually derived from data and reach explicatory value by relating certain parameter combinations to prevalent erosion- and transport processes. In its simplest form sediment transport capacity $C$ is at least dependant on accumulated discharge and local gradient $C = Q^m \times S^n$. Although the range of (m, n) combinations is broad, we assume the ranges, mentioned by Kirkby (1990, cited in Beven, 1996) to represent the underlying erosion- and transport processes (Fig. 2a). With the model provided by Kirkby (1971), the erosion processes of diffusive soil creep, rain splash, soil

wash, and advective river transport result in typical 1D hillslope profiles given by Eq. 8 and shown in Fig. 2b. The profiles reflect also the theory from Tarboton (1997) that within a catchment context hillslope processes can be attributed to convex profiles (more diffusive than advective erosion processes) and channels to concave profiles.

$$Z_{typ}(x) = Z_0 * \left( 1 - \left( \frac{x}{x_{HS}} \right)^{\frac{1-m}{1+n}} \right) \tag{8}$$

Eq. 8 is valid for the transport limited case and a hillslope with a close to constant width along the flow path.



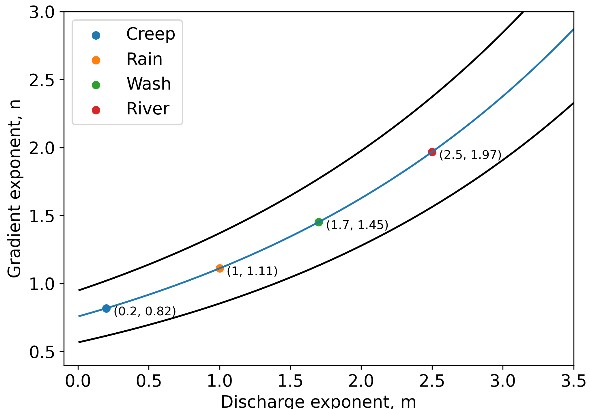 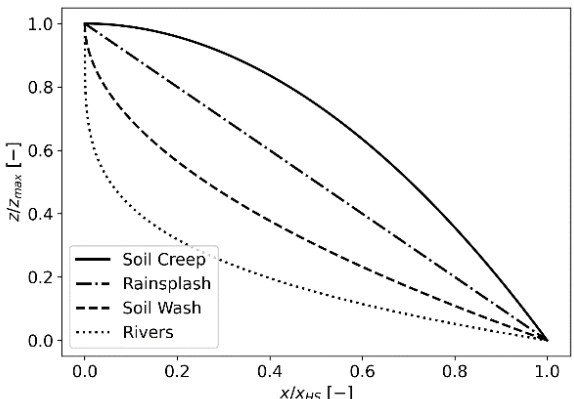

**Figure 2: a) Parameter ranges for typical erosion processes (Beven, 1996); b) Resulting 1D profiles for highlighted parameter combinations (cf. Fig. 2a)**

In our previous study we have already shown that convex profiles maximize dissipation of surface runoff per input flux of energy (precipitation) whilst also showing maximum rates of kinetic energy export at the downslope end. This is possible as kinetic energy is on a scale of 1000 times smaller than influx potential energy, therefore not significantly affecting the overall energy balance of a hillslope profile. In this context we extend this steady state analysis to account for the transient state of surface runoff and analyse maximum power and total work during a full surface runoff event. For simulation of these rainfall-runoff events, we implemented a solver of the 1D St. Venant equations for viscous flow and analysed the hydraulic variables on a space-time grid. The events were simulated for soil creep (SC) and soil wash (SW) profiles as their distributions of geopotential gradient show largest differences.

### 3.2 Numerical model for transient surface runoff

The simulation of surface runoff on 1D hillslope profiles, related to the erosion processes was done by numerical approximation of the system of equations, known as the shallow water equations. In this study we solve the conservative form of the 1D mass and momentum equations:

$$\frac{\partial X}{\partial t} + \frac{\partial F}{\partial x} = S \tag{9}$$

where

$$X = \begin{bmatrix} H \\ q \end{bmatrix} \qquad F = \begin{bmatrix} q \\ \frac{\beta q^2}{H} + \frac{gH^2}{2} \end{bmatrix} \qquad S = \begin{bmatrix} I \\ gH\frac{\partial z}{\partial x} - gq|q|\frac{n^2}{H^{\frac{7}{3}}} \end{bmatrix} \tag{9 (a, b, c)}$$

We applied a finite difference time variation diminishing (TVD) MacCormack scheme, which is presented in Liang et al. (2006). In this study we adjusted the source term by including the rainfall rate $I$ in m s$^{-1}$ and we approximated the friction term by the Manning-Strickler equation (Das et al., 2015) with the Manning coefficient $n$ in m s$^{-1/3}$ instead of the originally proposed



Chezy formula. $H$ is the total water column depth in meters, $g$ is the acceleration due to gravity (here 9.81 m s$^{-2}$) and $q$ the discharge per unit width in m$^2$ s$^{-1}$. $\beta$ is the correction factor for the non-uniform vertical velocity profile, which has been set to equal 1.0 for a uniform velocity distribution. Due to the influence of the water depth on the friction term, small and zero water

depths cause numerical instabilities and correct wetting-drying algorithms must be applied to insure stability of the numerical scheme (Liang et al., 2007). We applied similar to Vincent et al. (2001) an algorithm which sets the water depth during each computation time step to a minimum of 10$^{-5}$ m and no mass flux (q=0) at these points. The TVD term is included only at the inner computation points, excluding the boundary and the so-called ghost points, which are needed for the calculation of no boundary flux at the hillslope top (solid wall boundary) and the bottom outflow of the accumulated discharge (transmissive

wall boundary, cf. Causon and Mingham, 2010). In the following we briefly outline the MacCormack Scheme (MacCormack, 1969) with the additional TVD term (Liang et al., 2006) for Eq. (10c):

$$X_i^p = X_i^j - \left(F_i^j - F_{i-1}^j\right) * \frac{\Delta t}{\Delta x} + S^j * \Delta t \tag{10a}$$

$$X_i^c = X_i^j - \left(F_{i+1}^p - F_i^p\right) * \frac{\Delta t}{\Delta x} + S^p * \Delta t \tag{10b}$$

$$X_i^{j+1} = \frac{X_i^p + X_i^c}{2} + TVD(X_i^j) \tag{10c}$$

The superscripts $p$ and $c$ denote the predictor and corrector steps, while $j$ and $i$ represent the discretization in time and space. It is important to note that the spatial flux term $F$ is discretized backwards in the predictor time step and discretized forward in the corrector time step. The main benefits of this two-stage scheme are that one can solve regions with sharp gradients

through the inclusion of the TVD term and that the source term is computationally efficiently treated, whilst maintaining second-order accuracy, in time and space. The complete implementation of the scheme, including transmissive and solid wall boundary conditions is presented as python script in the supplemental code to this publication.

### 3.3 Averaging in time and space

Depending on the space and time discretization we can analyse how much of the energy influx by rainfall was converted into

free energy of overland flow and how much has dissipated. It is however not trivial to disentangle energy fluxes in space and time, and less so to analytically average over both domains to describe the nature of transient energy conversion rates. On the one hand, averaging over the time domain is typically accompanied by setting time derivatives to zero and allows us to analyse the steady state spatial distribution of energy (Schroers et al., 2022). On the other hand, averaging over the space domain leads to a black box system where we are unaware of the internal spatial distributions and only express the temporal evolution of the

system (e.g., Kleidon et al., 2013).

As the partial differential equations of the underlying movement of water (mass and momentum balances) are numerically approximated on a space-time grid, only an average of the energy fluxes in both domains provides an estimate for an entire hillslope and event. In this section, to introduce the reader to the general dynamics of transient surface runoff, we spatially

 

lump the entire hillslope into one OTS which is transient in time. In section 4 of this study, we extend this concept and double

average in space as well as in time. Fig. 3 shows the space-time grid, where at the computation points (circles) the hydraulic

variables $H$ and $q$ are calculated. An exemplary OTS is discretized in space with length $dx$ and temporal conversion dynamics

of energy for a time interval with length $dt$. Spatial derivatives of Eq. 4 $\left(J_{f,net}^{pe}, J_{f,net}^{ke}\right)$ are averaged in time (dt) and temporal

derivatives $\left(\frac{dE_{pe}}{dt}, \frac{dE_{ke}}{dt}\right)$ are averaged in space (dx), leading to a double averaging of power and dissipation of surface runoff

(Fig. 3). For calculation of space- and time derivatives between computation points $i$ $(j)$ and $i + 1$ $(j + 1)$ we apply forward

differencing, which reads $\frac{df(y)}{dy} = \frac{f^{i+1}(y) - f^i(y)}{dy}$, where y is the averaged variable in time $\left(\tilde{q}, \tilde{H}\right)$ or space $\left(\bar{q}, \bar{H}\right)$.

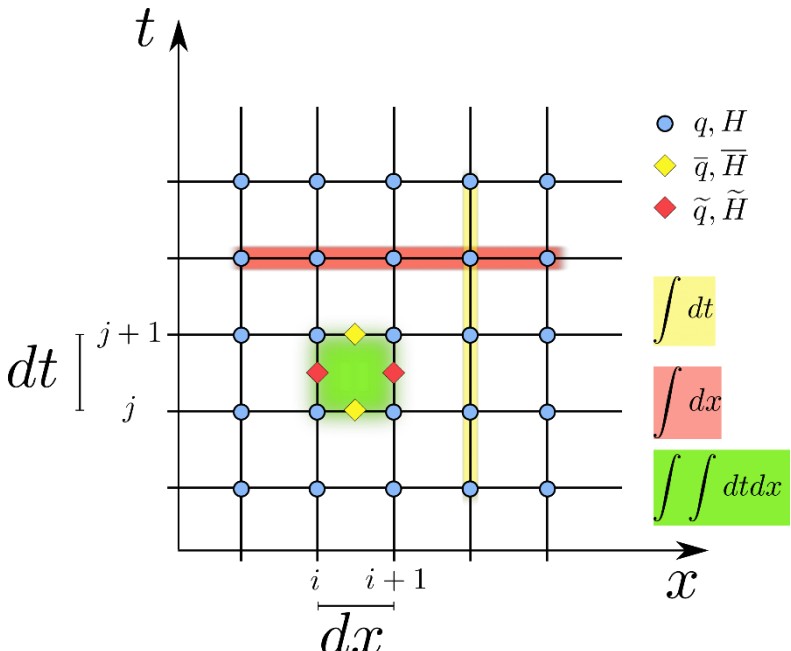

**Figure 3: Discretization of energy conversion dynamics in space (x) and time (t). $q$ and $H$ are evaluated on nodes (blue circles), energy conversion in time is integral over space (red) [W], in space is integral over time (blue) [J m⁻¹] and total energy converted is calculated as space-time integral (green) [J].**

Eq. 4 (Eq. A8 respectively, cf. Appendix A) forms the basis for an analysis of surface runoff in space and time. Depending on

the system and the rainfall-runoff event we define spatial and temporal boundaries to calculate the total converted energies.

For a defined OTS this allows for calculation of power and dissipation by integration: Either for the whole OTS (Fig. 3 red

area) in *W*, for the whole event (Fig. 3, blue area) in *J m⁻¹*, or for a specified duration and distance, averaged in time and space

(Fig. 3, green area) in *J*.




### 3.4 Scenarios and results

To highlight the different transient behaviours of characteristic hillslopes we compare the hillslope form which is related to advective soil wash erosion (SW) with the one which relates to diffusive soil creep (SC). We ran three simulation scenarios on each hillslope, differing in block rainfall rates (100 mm hr$^{-1}$ and 50 mm hr$^{-1}$) as well as length of rainfall time interval (120s

and 360s) (cf. Fig. 4). Based on the calculated hydraulic results we then proceeded to calculate the transient energy balance averaged in space over the hillslope length. Finally, the residual of the energy balance is interpreted as the total amount of dissipated energy in time and is analysed relative to the accumulated influx of energy by rainfall ($\widehat{D}_{HS}$, cf. Eq. 7), which allows a thermodynamic description of a temporally transient rainfall-runoff event.

### 3.4.1 Scenarios

The three analysed scenarios have been computed by the described numerical implementation of the 1D shallow water equations, the simulated hydrograph of each scenario is plotted in Fig. 4. In the first and third scenario (S1 and S3) both hillslope forms reach steady state (approximated as $\dot{Q} = 0$, if $\frac{\Delta Q}{Q} < 0.01$), where SW-hillslope forms reach steady state in less time than SC-hillslope forms. Scenario S2 describes a case without a steady state runoff regime. For all cases it is apparent that SC forms react faster to rainfall for the rising as well as the falling limb of the hydrographs. Interestingly, different rainfall

rates lead to different time intervals until the runoff can be described as steady state (cf. S1 and S3), with higher rainfall rates leading to a relative faster reaction of the hillslope and a longer interval of steady state runoff conditions. This relates to the nonlinear character of the simulated shallow water equations, as water accumulates faster on the surface, average runoff velocities grow as well.

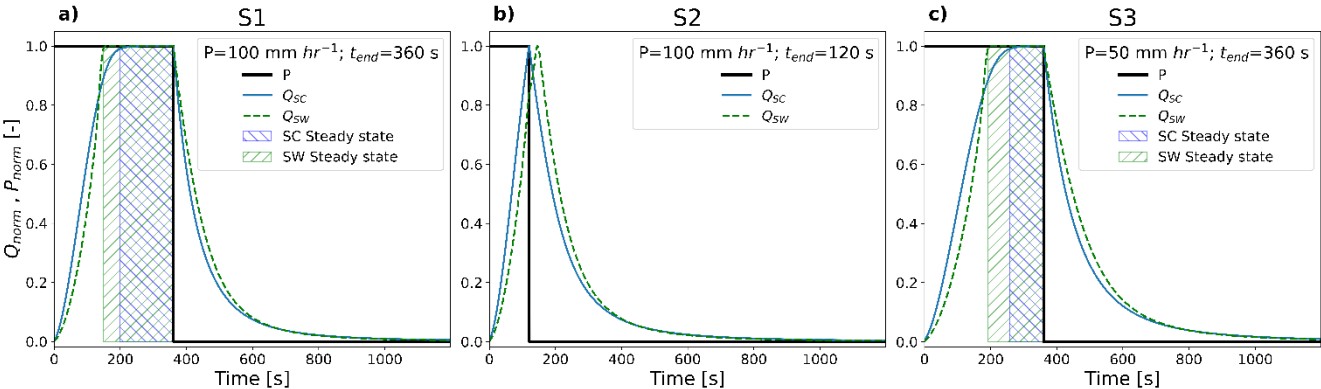


**Figure 4: Block rainfall scenarios and simulated hydrographs for SC- and SW- related 1D hillslope profiles**




### 3.4.2 Energy conversion dynamics

In the presented transient framework, an influx of energy may either lead to an increase of stored potential energy $\dot{E}_{HS}^{pe}$, an increase of kinetic energy $\dot{E}_{HS}^{ke}$, or an increase of the outflux of kinetic energy $J_{HS,out}^{ke}$ (Fig. 1). If these energy fluxes are positive

the energy is not dissipated and instead maintained as free energy of surface runoff. $\dot{E}_{HS}^{pe}$ and $\dot{E}_{HS}^{ke}$ contribute to the stored energy on the hillslope during the rising limb of the hydrograph, recede to zero when reaching steady state and dissipate during the falling limb of the hydrograph ($\dot{E}_{HS}^{pe} < 0$, $\dot{E}_{HS}^{ke} < 0$).

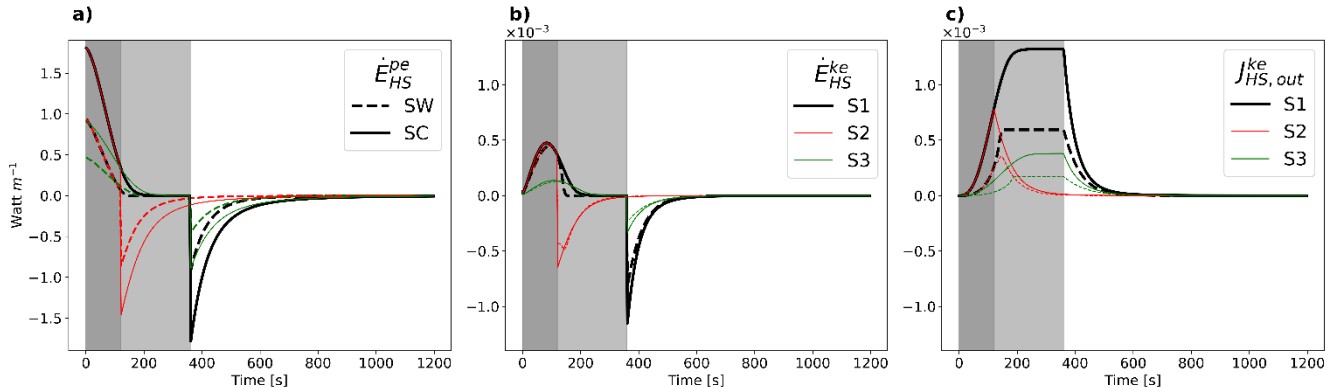

**Figure 5: Simulated temporal dynamics of spatially lumped a) stored potential energy; b) stored kinetic energy and c) kinetic energy**
**outflux in watt per meter flow width for SW- and SC- related 1D hillslope profiles.**

For all simulated scenarios the total energy which is stored and released is larger for SC than for SW profile forms (Fig. 5 a and b). The shortest interval to reach steady state is achieved for SW hillslopes and largest rainfall rates (S1), and contrarily the longest time interval for reaching steady state is related to SC hillslope forms and smallest rainfall rates (S3). Scenario S2 does not reach steady state runoff and follows the energy dynamics of S1 during the rising limb of the event (both have equal

rainfall rates). As however less energy has been stored on the hillslope for S2 than for S1, less energy is dissipated during the falling limb of S2 than of S1. For potential energy most energy is created at the beginning of the event, with small runoff depths and little to no flow. Most internal kinetic energy $\dot{E}_{HS}^{ke}$ is produced when flow depths rise (and therefore $\dot{E}_{HS}^{pe}$ falls) whilst the output of kinetic energy $J_{HS,out}^{ke}$ still hasn't reached it's maximum. $J_{HS,out}^{ke}$ is linked to the observed runoff at the downslope end of the hillslope profile and is for all three scenarios larger for SC than for SW hillslope forms (Fig. 5c). This export of

energy from the system is linked to the internal work from overland flow on the system, the longer it takes for the hillslope system to reach a steady state value of $J_{HS,out}^{ke}$ the more energy is available to perform work on the surface structures. This reflects our notion that certain hillslope morphologies are more likely to experience an overshoot in power and consequently more work which is generated by surface runoff.



### 3.4.3 Dissipation and energy efficiency

As outlined in the previous section, we approximate the dissipated energy integrated over the hillslope length as the energy residual of the computed hydraulic variables $q$ and $H$ (cf. Appendix A). The temporal evolution of dissipation $D_{HS}$ in watt per meter flow width for all simulated scenarios is plotted in Fig. 6a. In absolute terms dissipation rates are for each simulated scenario larger for SC than for SW hillslope forms. This result is independent of the transient temporal evolution of $D_{HS}$, maximum dissipation rates relate to the fully developed steady state and are at each point in time larger for SC than for SW

hillslope profiles.

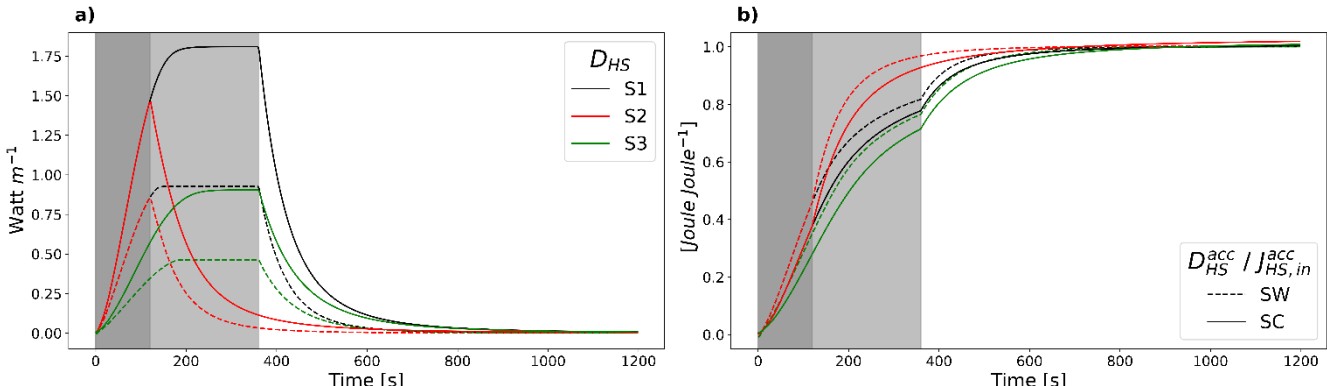

**Figure 6: Computed transient results of a) absolute dissipation $D_{HS}$ and b) relative dissipation $\widehat{D}_{HS}$ for scenarios S1, S2, S3 on hillslope profiles realted to soil wash (SW) and soil creep (SC)**

In this setup SW forms receive less influx of energy than SC forms and dissipation rates therefore need to be normalized by the influx of potential energy by rainfall to evaluate how much relative free energy is dissipated per hillslope type and scenario. We therefore computed $\widehat{D}_{HS}$, the fraction of accumulated dissipation $D_{HS}^{acc}$ per accumulated influx energy $J_{HS,in}^{acc}$ (Fig. 6b, cf. Eq. 7). This thermodynamic descriptor represents at each point in time the amount of energy which has already dissipated from the accumulated influx of free energy, a higher value means that friction is relatively larger and the runoff process less energy

efficient. At the end of the event this descriptor is close to 1 as almost all influx energy has dissipated at $t_{end} = 1200\,s$. In Fig. 6b we plotted relative dissipation $\widehat{D}_{HS}$ for the simulated hillslope profiles and scenarios. For all scenarios $\widehat{D}_{HS}$ is larger during the transient runoff event for SW than for SC hillslope profiles. This result is the opposite of the absolute values of dissipation and highlights the effect of normalizing energy conversion rates. Interestingly, larger rainfall rates (scenario S1) lead to larger relative dissipation rates than smaller rainfall rates (scenario S3). This means that although larger rainfall rates

lead to higher kinetic energy production $J_{HS,out}^{ke}$ (Fig. 5c), kinetic energy rates are much smaller than dissipation rates, allowing relative dissipation rates to be highest for largest rainfall rates and SW hillslope profiles. Scenario S2 without steady state runoff conditions leads to larger $\widehat{D}_{HS}$ values during the falling limb of the hydrograph, with a larger fraction of energy being dissipated at any point in time during the rainfall runoff event than for S1 or S3.





**3.5 Discussion**

In this first part of the study, we highlight the connection between surface runoff, dissipation of its free energy and the evolution of surface morphology. We argue in line with Wolman and Gerson (1978) and Beven (1981) that such events in nature are highly intermittent and transient in time, leading to the question how this can be interpreted within an optimality context such as has been proposed by many (cf. Singh, 2003, for an overview). Therefore, we put forward the concept of relative dissipation of free energy or equivalently energy efficiency of surface runoff, which is similar to Carnot's theorem of maximum work

which can be extracted from heat flow (Kondepui and Prigogine, 1952). This idea was applied to surface runoff on characteristic 1D hillslope profiles which are related to diffusive soil creep erosion and advective soil wash erosion. Interestingly our results show that the latter (SW) results in less energy efficiency of surface runoff, or differently stated a larger fraction of the provided free energy by rainfall is dissipated than for SC hillslope types (cf. Fig. 6b). This means that there is relatively more energy available for work on the surface of SC profiles (be it in the form of detachment or transport of

sediment particles). This reflects the generally accepted theory of the evolution of hillslope profiles (Kirkby, 1971) and river profiles (Leopold and Langbein, 1962) towards concave distributions of geopotential, e.g., a falling energy slope along the flow path. Although we do not specifically account for energy of sediment particles, we derive a simple starting point for a thermodynamic interpretation of erosion regimes and resulting geopotential distributions. The simulated scenarios also hint at the evolution of runoff response. If relative dissipation rates are analyzed on an event scale, our results show for the same

hillslope, shorter but more intense runoff events maximize relative dissipation and minimize energy efficiency.

We stress that these scenarios are only adequate for situations where infiltration is negligible as a loss of mass affects the transient energy balance. Furthermore, we did not touch small scale geomorphological adaptations such as rills. We showed in our previous study (Schroers et al., 2022) for steady state overland flow that rill processes are linked to the distribution of

dissipation rates and therefore affect the energy balance. The development of rills is however transient (Rieke-Zapp and Nearing, 2005) and reflects our notion that structural adaptations are a result of an overshoot of power. As a starting point it is therefore important to understand during which situations such an overshoot is more likely and transient structural adaptations will occur. The here proposed transient, event-based perspective highlights that larger rainfall rates and shorter rainfall overland flow events lead to larger relative dissipation rates- which is somewhat counterintuitive as flow velocities and kinetic

energy increase as well. The reason for this effect is that larger flow depths increase flow velocities and therefore facilitate during the transient state a faster depletion of the influx of potential energy through rainfall, while relatively less free energy is stored on the hillslope. In terms of energy efficiency of overland flow this means that long duration, small intensity rainfall overland flow events are most efficient, in contrast to short, high intensity rainfall overland flow events where a larger fraction of the provided free energy dissipates faster. Following this logic, structural patterns on hillslopes should organize over time

to decrease efficiency. This means that if we would apply the same event to a hillslope surface twice, the first event will produce smaller relative dissipation rates than the second. Simultaneously, the kinetic energy of surface runoff would increase



for the second event as the provided energy gradients are depleted faster. The latter coincides with the theory about minimization of energy expenditure (Rodriguez-Iturbe et al., 1992) as well as experimental results on the plot scale (Rieke-Zapp and Nearing, 2005).


This can also be explained with the maximum power principle (Lotka, 1922; Kleidon, 2016) which states that the open thermodynamic system organizes its internal structure to deplete the driving gradients at the maximum rate. In the case of runoff on a hillslope this would imply that given no other constraints, the hillslope erodes towards a configuration which reacts for the same rainfall event faster with larger runoff rates. The maximum power would be achieved once the runoff approaches

the shortest possible runoff response and largest runoff rate. Obviously, this is an extreme case which cannot be achieved in nature as geology, soil composition and vegetation constrain the runoff response, but this example helps to understand the evolution of the interaction between runoff and erosion. In the second part of this study, we build on these theoretical results, but extend the concept to real world hillslopes and observed runoff responses in the Weiherbach catchment and analyze whether erosion and the evolution of surface runoff is indeed linked to maximum power of surface runoff.

**4 Application to surface runoff events in the Weiherbach catchment**

Following our argumentation from the previous section, we apply the developed theory about energy efficiency of overland flow to observed rainfall runoff events in the Weiherbach catchment. The catchment has been subject to intensive monitoring which includes data about erosion and sediment transport, allowing in addition to overland flow for an analysis of erosion patterns within the presented energy efficiency framework.

**4.1 The Weiherbach catchment and the flash floods of 1994/1995**

The hilly Weiherbach catchment lies in the Kraichgau, which is in the south-west of Germany (Fig. 7). The latter has a size of 3.45 km$^2$ and has been a hydrological observatory of for more than three decades (Plate and Zehe, 2008). The result is a rich data set with multiple continuous time series of discharge, precipitation, climate parameters as well as soil humidity. Furthermore, several measurement campaigns yielded a spatially distributed set of soil hydraulic parameters (Zehe et al., 2001),

Manning-Strickler values of the principal land uses as a function of plant growth stage (Gerlinger, 1996), and annual cycles of morphological as well as physiological plant parameters. Sediment concentrations measurements at the two discharge measurement stations allowed balancing of total sediment loads (Scherer, 2008). Approximately ninety percent of the catchment is agricultural land use, of which the principal plant cultivations are wheat, corn, turnip and sunflower (cf. Fig. 7b).



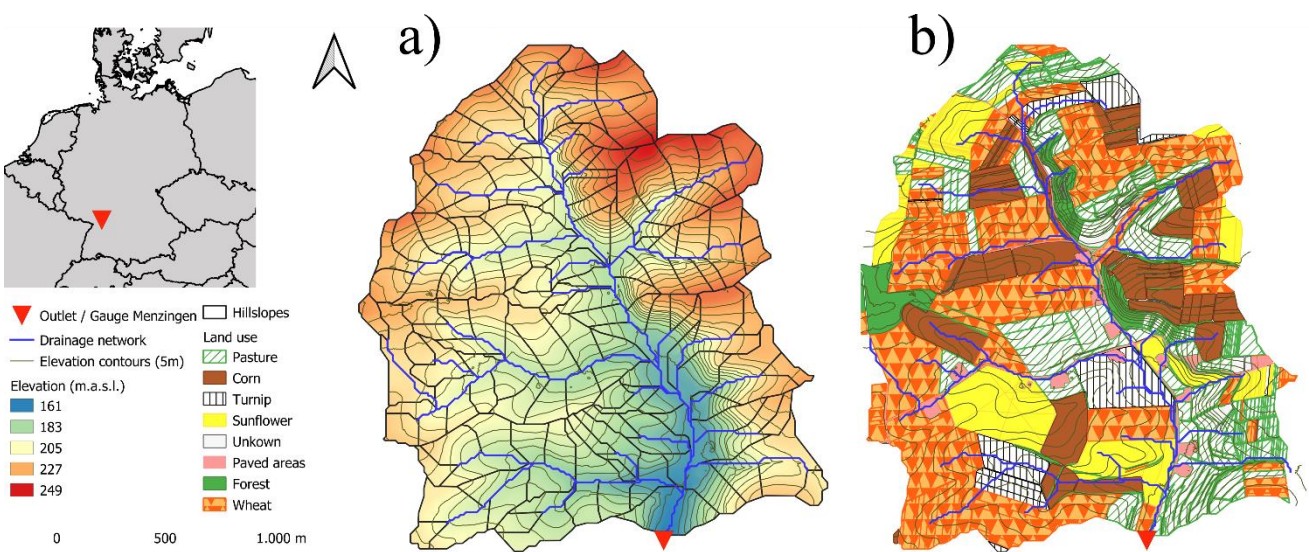

**Figure 7: The Weiherbach catchment: a) Observed drainage network, surface elevation and derived hillslopes (cf. Zehe et al., 2001); b) Land use patterns during the monitoring period (Scherer, 2008)**

The two largest runoff events were recorded on the 27th of June 1994 and on the 13th of August 1995 - in the following we will focus on these two events only and we will therefore refer to them as event 1 and event 2 or by year only (cf. table 1). Both events were caused by a convective precipitation event with a return period of 200a according to the KOSTRA data set

(Junghänel et al., 2010). However, the discharge peaks of both floods lie well above the 10000-year flood of 3.3 m³ s⁻¹ (BW-Abfluss (Blatter et al., 2007). A more detailed analysis of the event runoff generation can be found in Zehe et al. (2005), while for the study at hand we conclude that the recurrence intervals of peak discharge suffice to consider them effective in terms of landscape formation (cf. Beven, 1981), as corroborated by the considerable amounts of eroded sediments.

**Table 1: Hydrological variables for extreme events of 1994 and 1995**

| Event | Date | $I_{cum}$ [mm] | I [mm/h] | QP [m³/s] | RC [-] | $\bar{\theta}$ [m³m⁻³] | $T_I$ [a] | $T_{QP}$ [a] | $M_{sed}$ [t] |
|---|---|---|---|---|---|---|---|---|---|
| 1 | 27.06.1994 | 78,3 | 22,0 | 7,9 | 0,12 | 0,25 | 200 | $> 10^4$ | 1800 |
| 2 | 13.08.1995 | 73,2 | 23,0 | 3,2 | 0,07 | 0,26 | $> 100$ | $10^4$ | 500 |

**4.2 Model description and calibration**

The model we used is an extended version of the physically based model CATFLOW (Maurer, 1997; Zehe et al., 2001) which incorporates a sediment erosion module (Scherer, 2008). In brief, the model subdivides a catchment into several hillslopes and a drainage network, where each hillslope is discretized into a two-dimensional vertical grid. The widths of the elements vary from the top to the foot of the hillslope. For each hillslope, the model simulates the soil water dynamics and solute transport

based on the Richards equation in the mixed form as well as a transport equation of the convection diffusion type. The equations





are numerically solved using an implicit mass conservative Picard iteration (Celia et al., 1990) and a random walk (particle tracking) scheme. The simulation time step is dynamically adjusted to achieve an optimal change of the simulated soil moisture per time step which assures fast convergence of the Picard iteration. The hillslope module can simulate infiltration excess runoff, saturation excess runoff, lateral water flow in the subsurface and return flow. However, in the Weiherbach catchment

only infiltration excess runoff contributes to storm runoff and lateral flow does not play a role at the event scale. What is important is the redistribution of near surface soil moisture in controlling infiltration and surface runoff. As the portion of the tile drained area in the catchment is smaller than 0.5%, we didn't account for tile drains in the simulation. The here presented setup of the Weiherbach catchment is based on simulations and results from Zehe et al. (2005), who subdivided the catchment into 169 hillslopes in relation to land use and soil patterns (cf. Fig. 7a). The total soil depth represented by the model was 2m,

Manning roughness coefficients for the hillslopes and channels were taken from the mentioned experimental database (Gerlinger, 1997), while relative distribution of macroporosity at the hillslope scale was measured by Zehe (1999). The latter scales the total infiltration capacity during rainfall events in relative terms of the soil hydraulic conductivity, after the soil water content increases field capacity. The model was calibrated by stepwise increasing of macroporosity variability (Zehe et al., 2005) for event one and two (table. 1), yielding Nash-Sutcliff model efficiencies of 0.97 (event 1) and 0.98 (event 2) at the

downstream gauge in Menzingen (Fig. 8a and Fig. 8b). The main storm runoff generation mechanism for both events is infiltration excess runoff, which is routed in the model on the hillslopes into the channel, both based on the advection-diffusion approximation to the one-dimensional Saint-Venant equations. Individual surface runoff responses of each hillslope $Q_{HS}^i$, mean of all hillslopes $Q_{HS}^m$ and for both events can be seen in Fig. 8. For reasons of briefness, we refer to Maurer (1997) or Zehe et al. (2001) and Zehe et al. (2005) for more details on model structure and model equations, as well as the parameters

of the river network.

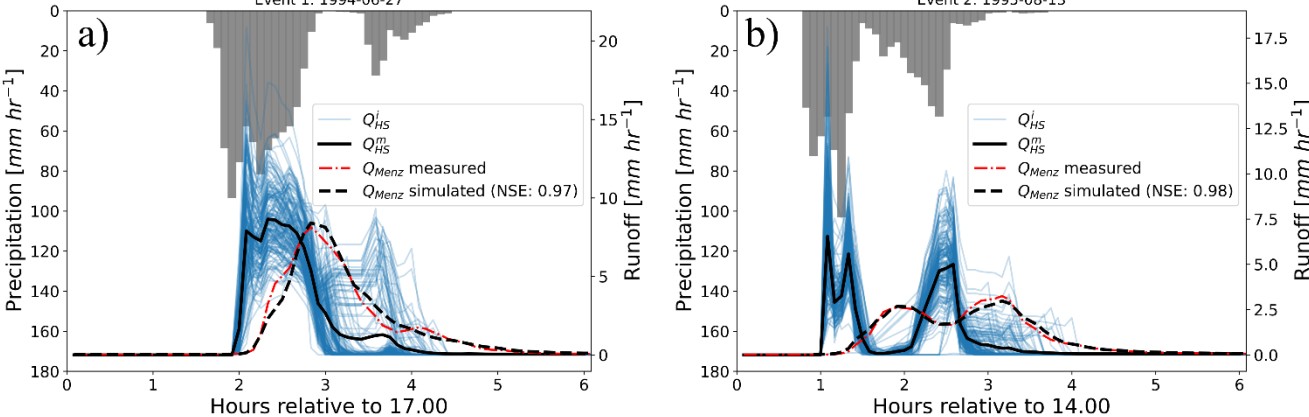

**Figure 8: Observed precipitation and catchment discharge response, simulated surface runoff at hillslope scale $Q_{HS}$ as well as simulated river discharge $Q_{Menz}$ for a) the event of 1994-06-27 and b) the event of 1995-08-13.**

Sediment erosion and transport is modelled using the steady state sediment continuity equation (Eq. 12). Sediment transport

capacity follows an adjusted concept from Meyer and Wischmeier (1969), treating sediment detachment and transport as





individual processes. Potential erosion $e_{pot}$ (kg m$^{-2}$ s$^{-1}$) is simulated in CATFLOW-SED (Scherer et al., 2012) by a semi-empirical approach that bilinearly accounts for detachment by rainfall momentum flux $m_r$ (N m$^{-2}$) as well as overland flow shear stress $\tau$ (N m$^{-2}$) (cf. Eq. 11).

$$e_{pot} = p_1 \cdot (\tau + p_2 \cdot m_r - f_{crit}) \quad if \ e_{pot} < 0, e_{pot} = 0 \tag{11}$$

The resisting forces acting against detachment are characterised by two empirical parameters: the erosion resistance $f_{crit}$ (N m$^{-2}$) as well as the erodibility parameter $p_1$ (-), scaling the growth of the detachment rate in case the attacking forces exceed the threshold $f_{crit}$. The parameter $p_2$ (-) weighs the momentum flux of rainfall against shear stress from overland flow. The empirical parameters were determined for conventionally tilled loess soils using data from rainfall simulation experiments performed in the laboratory (Schmidt, 1996) and at erosion plots in the field (Scherer et al, 2012). Sediment transport is modelled with the approach from Engelund and Hansen (1967) empirically relating a dimensionless transport intensity to dimensionless stream

intensity and consequently allowing for a calculation of transport capacity based on hydraulic overland flow conditions. Sedimentation of suspended particles is accounted for depending on Reynolds number and the particle size, characterizing their buoyancy. At each timestep CATFLOW-SED then balances sediment transport for each overland flow element based on the stationary form of the sediment continuity equation (Eq. 12).

$$\frac{\partial q_s}{\partial x} = \Phi(x, t) \tag{12}$$

Where $q_s$ is sediment mass flow per unit width in kg m$^{-1}$ s$^{-1}$, $\Phi$ net detachment/ sedimentation of sediments from overland flow

in kg m$^{-2}$ s$^{-1}$, $x$ length coordinate in meter and $t$ time step in seconds. For more details on the implementation and model equations we refer to Scherer et al. (2012) and Scherer (2008). The sediment transport model was able to simulate total erosion for both flash floods with an absolute error of 8% (see table 1), which is within the error margin of the observations. As previously mentioned, deposition and erosion patterns for individual hillslopes indicate that especially convex shaped slopes with highly erodible crop types result in high erosion rates (Fig. 7). In the Weiherbach catchment these slope types are located

in the east.

### 4.3 Transient energy and power

### 4.3.1 Surface runoff

We estimated for both events the evolution of potential and kinetic energy on each hillslope as well as the kinetic energy export from the hillslope (cf. Eq. 6 and Eq. 7). $\dot{E}_{HS}^{pe}$ makes up by far the largest portion of free energy at any point in time, while $\dot{E}_{HS}^{ke}$

and $J_{HS,out}^{ke}$ can be considered negligible for the hillslope energy balance (cf. Fig. 9). For the event in 1994 $\dot{E}_{HS}^{pe}$ shows three positive and three negative peaks with very limited periods of time independence at roughly 2.5 h to 3.3 h (Fig. 9a). For $\dot{E}_{HS}^{pe}$ as well as $\dot{E}_{HS}^{ke}$ positive values represent an increase of free energy that is stored on the hillslope and thus an overshoot in power, while negative values indicate that stored free energy is decreasing (Fig. 9a and 9b). In contrast to the internal free energies, $J_{HS,out}^{ke}$ increases on average to a certain level and maintains this flux until the end of the rain event (Fig. 9c). From an external




perspective the system therefore seems to reach steady state but is internally in a transient unsteady state. At this stage it is

interesting to mention, that Zehe et al. (2013) quantified the power in soil water fluxes during these events and evaluated their

dependency on macroporosity, which resulted in values of 1-2 watt m$^{-2}$ per hillslope. This translates with a mean hillslope area

of 20000 m$^2$ into approximately 2-4 x 10$^4$ watt per hillslope, which is of the same scale as the sum of the here presented free

energy fluxes $\dot{E}_{HS}^{pe}$, $\dot{E}_{HS}^{ke}$ and $J_{HS,out}^{ke}$.

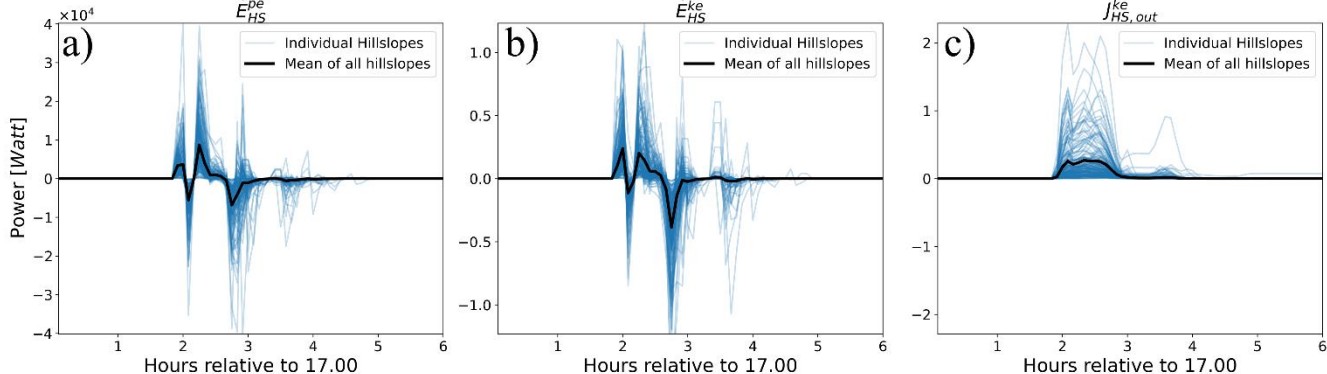


**Figure 9: Calculated free energy dynamics for the surface runoff event 1 on 1994-06-27 of changes in a) potential energy $\dot{E}_{HS}^{pe}$, b) kinetic energy $\dot{E}_{HS}^{ke}$ and c) energy out flux $J_{HS,out}^{ke}$**

Event 2 in 1995 (Fig. 10) shows similar energy dynamics but with lower magnitude and lesser maximum runoff rates. The

maximum peak of $\dot{E}_{HS}^{pe}$ is not mirrored by a negative counterpart (Fig. 10a), indicating that large amounts of stored surface

water infiltrates rather than contribute to further surface runoff. Its effect can also be seen from the dynamics of $J_{HS,out}^{ke}$ (Fig.

10c) which has on average 3 peaks with a dip in power between peak one and two, although energy influx from rainfall is

maintained almost constant during this period (cf. Fig. 8b).

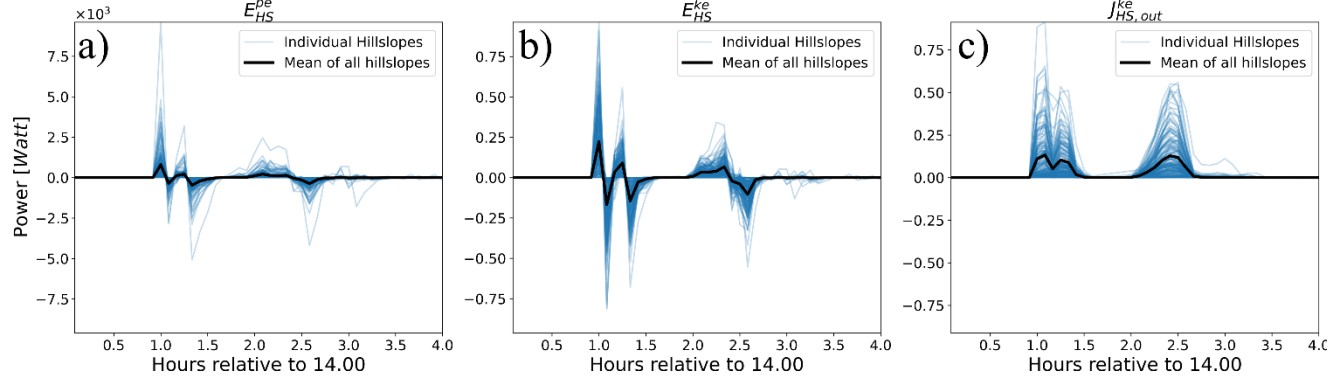

**Figure 10: Calculated free energy dynamics for the surface runoff event 2 on 1995-08-13 of changes in a) potential energy $\dot{E}_{HS}^{pe}$, b)**
**kinetic energy $\dot{E}_{HS}^{ke}$ and c) energy out flux $J_{HS,out}^{ke}$**



Using the energy influx $J_{HS}^{in}$ we calculated $D_{HS}$ for each hillslope (Eq. 5), event, and as average of all profiles (Fig. 11). $D_{HS}$ is very dynamic and is for both events unsteady, with a global maximum occurring at the beginning of an event and followed by one or more subsequent smaller local maxima. We also note that the spread of $D_{HS}$ between individual hillslopes is large, especially at the points in time of maxima.

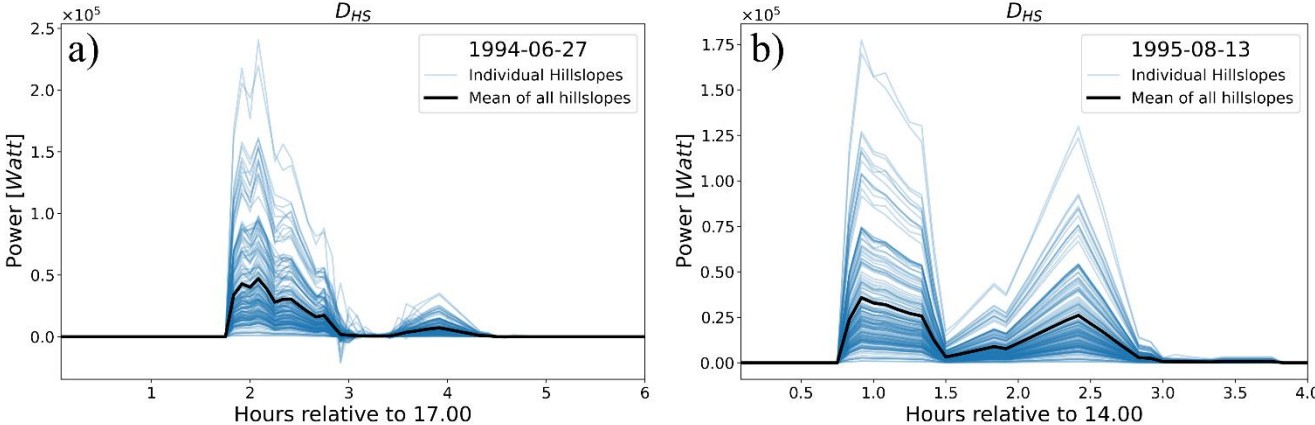


**Figure 11: Temporal dynamics of dissipation $D_{HS}$ for individual hillslopes and mean of all hillslopes for a) event 1 and b) event 2**

**4.3.2 Sediment transport**

For both simulated events the model was able to reproduce observed total sediment transport at the gauge Menzingen (cf. table 1). To estimate the average work of overland flow on sediments we analyse the accumulated spatial erosion- and deposition
patterns on each hillslope at the end of both events. We approximate the average kinetic energy that would be necessary to transport a given mass of sediment $m_{sed}$ (kg) for a representative length $l_{rep}$, which represents the average distance a sediment particle was transported during the time interval of overland flow $t_{sed}$. We calculate $l_{rep}$ by weighting of the downslope distance of each computation segment $s$ to the hillslope end with its related eroded or deposited sediment mass $m_{sed,s}$ in kg (Eq. 13). The sum of eroded and deposited sediment over all hillslope segments results in total eroded mass per hillslope $m_{sed,HS}$.

$$l_{rep} = \frac{\sum_{s=1}^{s_{end}} l_s * m_{sed,s}}{\sum_{s=1}^{s_{end}} m_{sed,s}} \tag{13}$$


The time interval during which overland flow was acting on bed material $t_{sed}$ was calculated from simulation results of each hillslope as the period of overland flow with mean overland flow depths larger than 1 mm. Total expended energy per unit area $e_{sed,HS}$ ($J\ m^{-2}$) is finally calculated for each hillslope as:

$$e_{sed,HS} = \frac{1}{2} * \frac{m_{sed,HS}}{A_{HS}} * \left(\frac{l_{rep}}{t_{sed}}\right)^2 \tag{14}$$




Where $A_{HS}$ is the hillslope area in $m^2$ and $m_{sed,HS}$ the eroded sediment mass in kg. Fig. 12a shows the simulation results for accumulated erosion per hillslope segment after the 1994 event (cf. Scherer, 2008). Negative values represent areas of deposited sediment whereas positive values indicate the erosion of soil. Erosion was large on highly erodible soils with little plant coverage such as sunflower or corn fields (cf. Fig. 7b). A difference between convex and concave hillslope profiles was visible, as the former allow for deposition of sediment at the hillslope feet due to a declining topographic gradient. Note that

hillslope form is incorporated in the estimated average expended energy on sediments as negative erosion (mostly deposition at the hillslope foot) reduces $l_{rep}$ (cf. Eq. 13) and $e_{sed,HS}$ (cf. Eq. 14). $e_{sed,HS}$ therefore, not only reflects the influence of soil erodibility due to land use and soil characteristics, but also implicitly informs on driving geopotential gradients. This can be seen by comparing the spatial patterns of erosion (Fig. 12a) and related expended energy $e_{sed,HS}$ (Fig. 12b): While absolute erosion rates are seemingly randomly scattered throughout the catchment, $e_{sed,HS}$ is clearly largest on the eastern slopes of the

catchment and to a lesser extent present on the western slopes (Fig. 12b). In the following we will make use of this information about geopotential gradients and analyse the east-west pattern with respect to energy efficiency of overland flow.

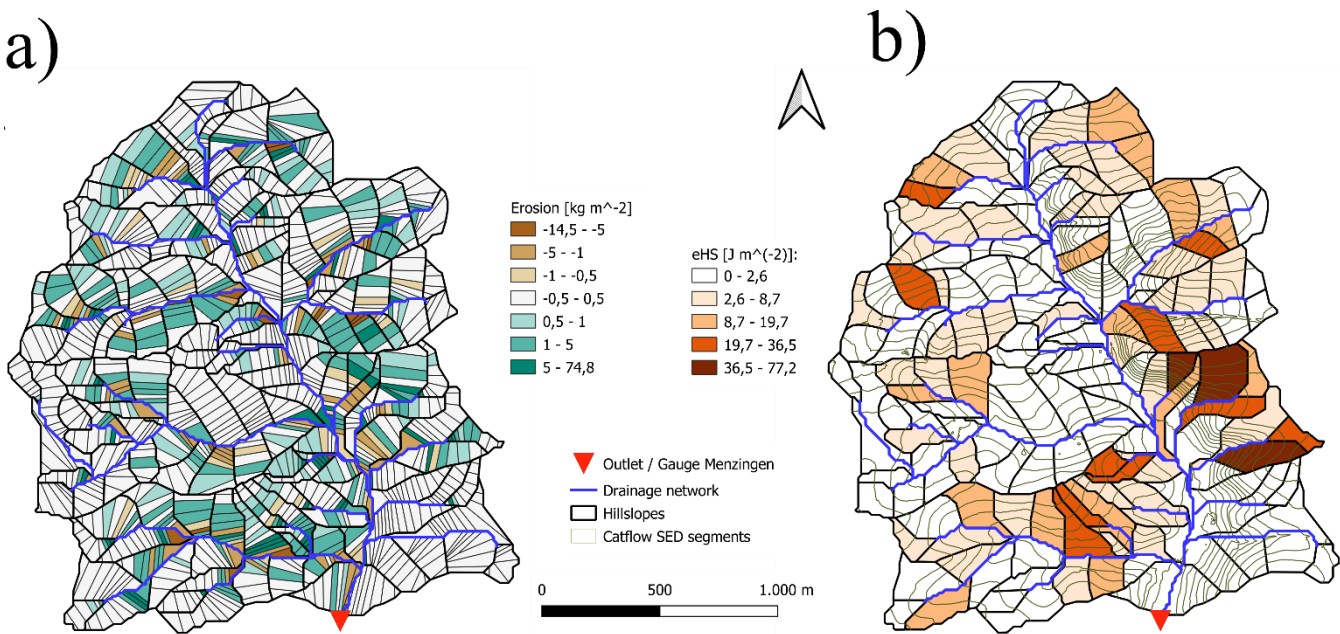

**Figure 12: a) simulated erosion and b) approximated expended energy on erosion per hillslope for the 1994 event**

**4.4 Energy efficiency of characteristic hillslope forms**

The calculations of transient energy and power for both calibrated rainfall runoff events provide an estimate of energy efficiency of overland flow for each hillslope in the Weiherbach catchment. These energy efficiencies are linked to the geomorphological development stage of each hillslope, facilitating an interpretation of geomorphology within the energy balance of surface runoff. To this end, we cluster the hillslopes into groups, representing the typical hillslope profile groups SW, RS, and SC, as introduced in sect. 3.1 and detailed below.





### 4.4.1 Clustering hillslope forms

To cluster the 169 hillslope profiles, each one is normalized in its vertical and horizontal length and then plotted as a single point into a three-dimensional space, consisting of the axis: 1) Mean vertical height, 2) Percentage length of negative curvature, 3) Horizontal length coordinate of maximum slope. The same procedure is applied to the normalized characteristic hillslope profiles SW, RS and SC from sect 3.1 forming cluster centroids. This allows clustering of model hillslopes according to their minimum Euclidian distance in the parameter space and resulted in 27 hillslopes being classified as SC type, 129 profiles as RS type and 13 belonging to SW (Fig. 13a). This confirms the perception that most erosion can be attributed to a combined impact of kinetic energy by rain splash plus shear stress of overland flow accumulation. The classification also showed that 27 hillslopes which can be related to soil creep lie mostly in the eastern part of the Weiherbach catchment (cf. Fig.13b), where highest erosion rates were simulated. In the next section we do not only confirm this general erosion pattern but also show that highest erosion rates coincide with highest relative dissipation rates and therefore maximum work which overland flow performed on the sediments.

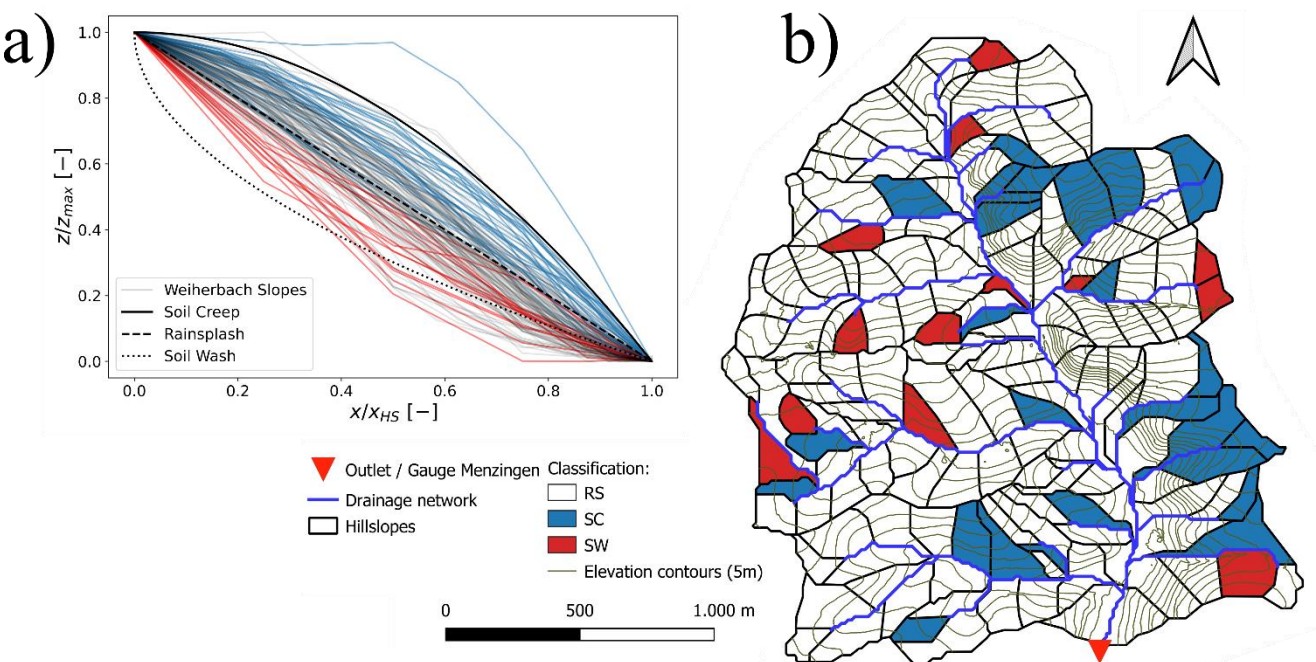

**Figure 13: Classification of Weiherbach hillslope profiles into forms related to soil creep (blue), rain splash (a) grey, b) white), and soil wash (red).**



### 4.4.2 Relative dissipation patterns and energy efficiency of surface runoff

For both events we plot the hillslope clusters for calculated total dissipated energy as well relative dissipated energy (Fig. 14).

In both cases we find distinct differences between SC, RS and SW hillslope types. In absolute terms, more energy is dissipated

for both events on SC profiles than RS and SW types, while SW types show lowest dissipated energy levels. Contrarily, relative

dissipated energy is highest for SW hillslope types and lowest for SC classified profiles. $\widehat{D}_{HS}$ values ranging from 91% to 99%

indicate that almost all energy has been dissipated or has been transferred to the sediments at the end of the rainfall event at

$t_{end}$=5h.

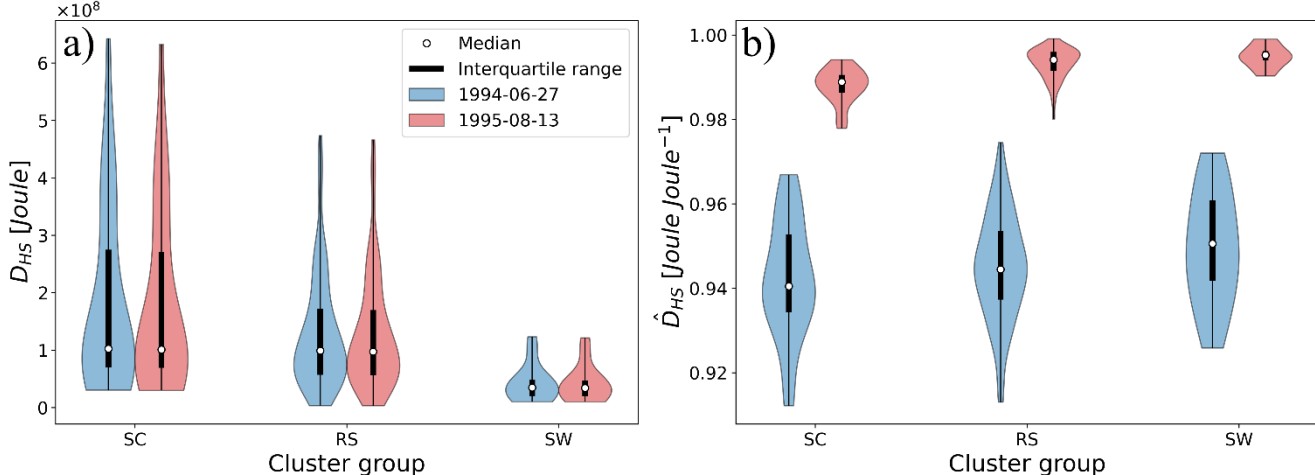

**Figure 14: Clusters of geomorphological hillslope types (SC, RS, SW) and a) dissipated energy $D_{HS}$ as well as b) relative dissipation $\widehat{D}_{HS}$ for runoff events 1 and 2**

SC hillslopes receive larger quantities of energy influxes through rainfall but in comparison to SW profiles dissipate a smaller

portion of this energy. Both events show similar total dissipated energy levels, which is due to very similar total rainfall

volumes. Fig. 14b however shows that although total energy influx and dissipation is similar, relative dissipation is larger for

the 1995 event than for the 1994 event. This difference arises from the larger surface runoff rates of the latter (due to less

infiltration (cf. Zehe et al., 2005) at its peak up to three times larger, cf. Fig. 8), leading to more kinetic energy of surface

runoff at the outlet.

Similarly, we compare relative free energies $\hat{E}_{HS}$ and relative outflux energies $\hat{J}_{HS,out}$ of the three hillslope types (Fig. 15).

Fig. 15a shows the maximum values of transient relative free energy that is not dissipated during each surface runoff event for

all simulated hillslopes (cf. Eq. 7b). The results indicate a tendency of SW and RS profiles to lead to less relative free energy

in comparison to SC hillslope profiles. Relative free energy $\hat{E}_{HS}^{max}$ mirrors $\widehat{D}_{HS}$, highlighting the connection between maximum

free energy that is stored in time on the hillslope and total dissipated free energy over the whole event.





Compared with each other, the 1994 event generates larger relative kinetic and potential energy fluxes than the 1995 event,
with less total runoff volume. $\hat{E}_{HS}$ of the 1994 event is therefore much larger than during the 1995 event.

Free energy during a transient event consists of the stored potential and kinetic energy as well as the energy outflux at the
hillslope end. An analysis of the latter (Fig. 15b) reveals that there is only a small difference between the three hillslope types
and between events. This means that for the analysed events hillslope geomorphology seems not to be imprinted in kinetic
energy export at the hillslope outlet.

These findings imply that during a surface runoff event, the largest differences between hillslope types can be observed in the
pattern of free energy components along the flow paths, and not locally, e.g., at the hillslope end. These results differ from our
previous analysis of steady state runoff, where SW hillslope types increased the relative kinetic energy outfluxes in comparison
to RS and SC profiles (Schroers et al., 2022). As the latter did not account for infiltration processes, we hypothesize that
distributed infiltration in the catchment levels out these differences.

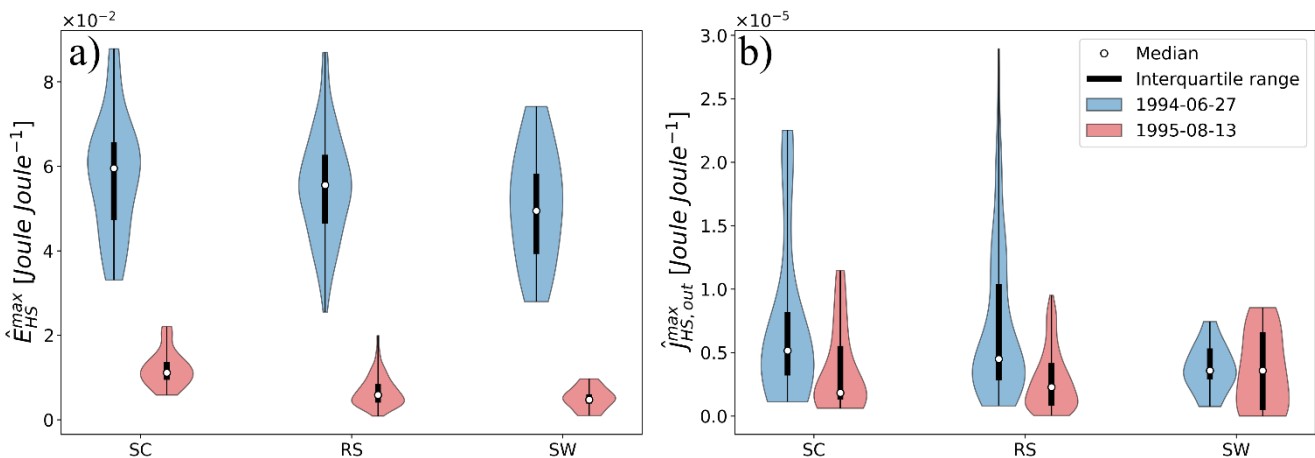


**Figure 15: Clusters of geomorphological hillslope types (SC, RS, SW) and maximum a) relative stored free energy $\widehat{E}_{HS}$ as well as
b) relative free energy flux $\hat{J}_{HS,out}$ at the hillslope foot for runoff events 1 and 2**

### 4.4.3 Erosion patterns

Mean erosion rates $e_m$ ($kg\ m^{-2}$) and accumulated erosion $e_{tot}$ (tonnes) for both events have been calculated by summing total
sediment- erosion and deposition of each hillslope. Similarly, we calculated the runoff coefficient $RC_{HS}$ of overland flow for
each hillslope. For the 1994 event $e_{tot}$ ranges between 0 to 90 tonnes per hillslope and $RC_{HS}$ lies between 0.05 and 0.52, while
for the 1995 event the corresponding ranges are 0-45 tonnes for $e_{tot}$ and 0.02 to 0.16 for $RC_{HS}$ (cf. Fig. 16). While there is no
correlation between these variables for neither of both events, we find a clear relation to the hillslope profile type. For both
events eroded sediment is smallest for profiles related to soil wash (SW) and largest for SC type profiles. Note that for the
1994 event the averaged eroded sediment per hillslope profile type $e_m$ is smallest ($e_m = 1.4$ tonnes) for SW, intermediate (10.2
tonnes) for RS and largest (23 tonnes) for SC profile (Fig. 16a). The same pattern is observed for the event of 1995 (Fig. 16b).
$e_{tot}$ on SW profile types accounts for only around 1% (18 tonnes) of total erosion in the catchment during the 1994 event and





3% (20 tonnes) during the 1995 event. Interestingly, the largest difference of eroded sediment between both events is observed on SC and RS profiles while mean as well as total eroded sediment of SW profiles is almost equal for both events. With respect to total runoff volumes, the results convey, that hillslopes with runoff coefficients in the medium range determine almost the entire erosion. For the event of 1994, hillslopes with $0.06 < RC_{HS} < 0.17$ account for 92% of total eroded sediment mass, while for the event of 1995, 95% of eroded mass occurred on hillslopes with $0.042 < RC_{HS} < 0.095$. Above and below these ranges none to very little erosion occurred. It is also noteworthy that for both events not only largest amounts of eroded sediment coincide with medium range runoff coefficients, but also that most hillslopes operate in this range.

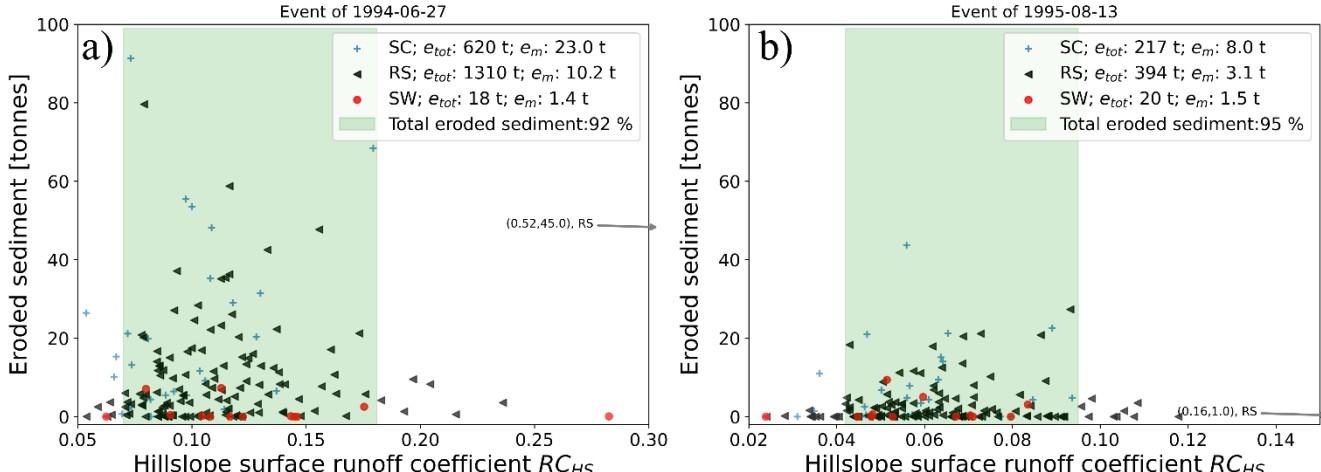

**Figure 16: Simulated surface runoff coefficient $RC_{HS}$ vs. eroded sediment for each hillslope of a) event in 1994 and b) event in 1995 for each hillslope and hillslope cluster (SC, RS, SW)**

For both events we then computed relative dissipation of overland flow and plotted the result against average expended energy on sediment transport per unit area for each individual hillslope (Fig. 17). We highlighted the medium ranges of relative dissipation $\widehat{D}_{HS}$ and kinetic energy of the sediments $e_{HS}$ for each hillslope cluster with kernel colour coding, which indicates a hierarchal structure of expended energy on sediment transport: $e_{HS}$ decreases from SC- to RS- to SW profile types. This marked difference can be seen for both events (Fig. 17a and b) and is highlighted by the mean expended energy on sediment transport per cluster group $e_{HS,m}$. Relative dissipation is as expected for both events and all hillslopes close to one, which suggests that most input energy is dissipated during the runoff process. Mean relative dissipation $\widehat{D}_{HS,m}$ is generally smaller for the 1994 event than for the 1995 event where less overland flow occurred. For both events $\widehat{D}_{HS,m}$ increases with changing hillslope type from SC to RS to SW, but this hierarchy is more pronounced for the 1995 event. We conclude that the results indicate a clear pattern of relative dissipated energy of overland flow and expended energy on sediment transport: On average, from SC, to RS, to SW $\widehat{D}_{HS}$ increases and $e_{HS}$ decreases. In plain words, if relatively more energy of the influx energy is





dissipated, less energy is available for erosion and sediment transport. A decrease of energy efficiency (equals increase of $\widehat{D}_{HS}$)

in overland flow is therefore related to a decrease of expended energy on sediment transport.

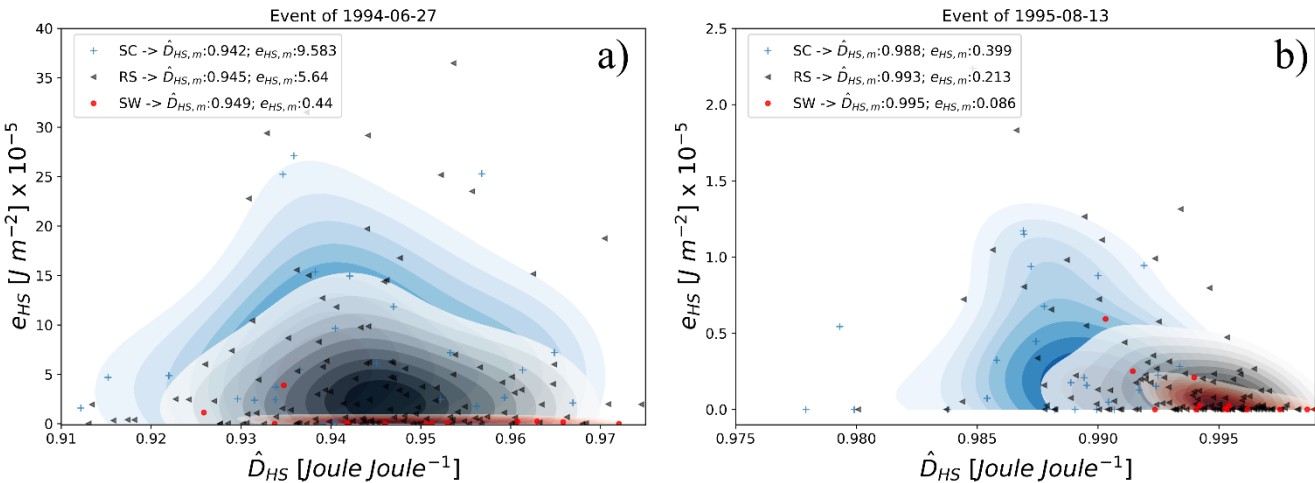

**Figure 17: Relative dissipated energy $\widehat{D}_{HS}$ vs. total expended energy for sediment erosion $e_{HS}$ of a) the event in 1994 and b) the event in 1995 for each hillslope and hillslope cluster (SC, RS, SW)**

**4.5 Discussion**

In this second part of the study, we have explored a range of concepts to connect runoff generation process, erosional regimes, and geomorphological evolution of hillslopes in a thermodynamic framework. We put the focus on the analysis of two extreme rainfall runoff events, which were observed in the Weiherbach catchment. This certainly raises the question how representative these events are- given their rare occurrence. We argue however, in line with Wolman and Gerson (1978) and also Beven

(1981) that only certain events contribute to effective landscape formation. Those events must be extraordinary as an overshoot in power is needed to exceed a threshold and trigger significant erosion and structure formation (Zehe and Sivapalan, 2009). Our analysis of the surface runoff during these two extreme events clearly shows, that driving downward, dissipative cascade of energy conversions from potential energy to kinetic energy and work on the sediment should be seen within a transient framework, as neither the mass nor the momentum balance during overland flow events is at steady state. We found that the

resulting power of surface runoff is of the same order as power of water infiltration into the soil via macropores (cf. Zehe et al., 2013). This might imply that surface and subsurface flow coevolve into a maximum power state where dissipation and power are equally distributed between complementary domains or more precisely flow paths (cf. Schroers et al., 2022).

We then connected the energy balance and energy efficiency of surface runoff events to the geomorphological forms of the derived hillslope systems. While this rests on the assumption that the delimited hillslopes represent homogeneous dynamics,

we are confident that this is the case as those are defined by topography as well as land use, the main controls of infiltration rate and surface runoff (Zehe et al., 2001). Most hillslopes were classified as profiles relating to rain splash erosion, and only few as soil creep or soil wash profile. We find a clear hierarchy relating relative dissipation, thus energy efficiency to erosion



rates. $\widehat{D}_{HS}$ is largest on SW then RS and smallest on SC profiles, indicating that SW profiles are conserving the least percentage of the energy influx by rainfall while SC profiles are most efficient in generating power in surface runoff. The energy efficiency

of overland flow $1\text{-}\widehat{D}_{HS}$ therefore constrains the effectiveness of a rainfall runoff event to change land forms and trigger landscape evolution (cf. Wolman and Miller, 1960). A larger value indicates that more potential energy is conserved as free energy, which implies that overland flow acts with larger average forces and can perform more work on the surface materials (overshoot in power for structure formation). The 1995 event on average resulted in larger $\widehat{D}_{HS}$ values than the 1994 event, which explains the higher erosion rates of the latter (cf. table 1: $M_{sed}^{1994} = 1800\ tonnes$ vs. $M_{sed}^{1995} = 500\ tonnes$ ).

Importantly, as accumulated rainfall amounts are almost equal for both events (cf. table 1: $I_{cum}^{1994} = 78.3\ mm$ vs. $I_{cum}^{1995} = 73.2\ mm$), this difference relates not to differences in energy influxes by rainfall. This is indicated by almost equal absolute dissipated energy (Fig. 14a) and can also not be deduced from kinetic energy fluxes at the hillslope feet (Fig. 15b). The difference between both events arises from storage rates of free energies within the hillslope systems in the form of potential and kinetic energies. Importantly, energy storage and therefore effectiveness of a surface runoff event relate to transient

conditions. Although we found that runoff coefficients and total erosion amounts were not correlated, largest erosion rates were found for hillslopes with a medium $RC_{HS}$. This is somewhat surprising as one would think that highest $RC_{HS}$ values would also result in largest erosion rates. However, our results give evidence that larger $RC_{HS}$ values are related to hillslope profiles which are closer to a dynamic equilibrium, store less free energy and therefore produce less erosion. The maximum work surface runoff can perform on the sediments relates to the potential flux in overland flow and thus on runoff and the

specific geopotential gradient (cf. Schroers et al., 2022). As the concept of relative dissipation captures both, we found a strong relation between mean $\widehat{D}_{HS}$ and the average work / free energy expended on sediments $e_{HS}$ (detachment and transport) for the three analyzed hillslope classes (Fig. 17). Clearly most work on the eroded sediments was performed on SC- and RS- and only very little on SW hillslopes. In terms of efficiency, we find that SC profiles are on average more efficient in power generation of surface runoff ($1 - \widehat{D}_{HS}$ is larger), which implies that more work can be performed on sediments ($e_{HS}$ is larger), while SW

profiles are less efficient ($1 - \widehat{D}_{HS}$ smaller, $e_{HS}$ smaller).

This finding is in line with a general pattern, characterizing the co-evolution of surface runoff dynamics, erosion and hillslope geomorphology, which holds for various climatological as well as geological settings (Perron et al., 2009). More generally, the evolution of the hillslope system towards less energy efficiency is consistent with the idea of maximization of dissipation and therefore entropy production (cf. Leopold and Langbein, 1962).

**5 Summary and Conclusion**

In this study we established a connection between morphological hillslope forms and their efficiency to power generation of overland flow from the energy input during rainfall events. We expanded the thermodynamic framework relating the steady state free energy balance of surface runoff to hillslope forms and the presence/ absence of a rill network (Schroers et al., 2022) to a) transient conditions and b) included the expended energy/ work performed on erosion and sediment transport. Releasing



the steady state assumption, essentially implies that the free energy balance of surface runoff, which constrains the maximum work surface runoff can perform on the sediments, relates to slope, form and structure of the hillslope and at the same time to the "refuelling" of the open system with potential energy during rainfall events. To account for both factors, we introduce the concept of relative dissipation, relating frictional energy dissipation to the energy input, which characterises energy efficiency of the hillslope when treated as open, dissipative power engine. We explored the transient free energy balance in terms of its

energy efficiency, comparing typical hillslope forms, representing a sequence of morphological stages and related dominant erosion processes (Kirkby, 1971.)

A first analysis, based on simulated synthetic events, suggested that older hillslope forms, where advective soil wash erosion regimes dominates, are less energy efficient in generating power during overland flow events, when compared to younger forms with diffusive erosion regimes. In the time domain we found that shorter, more intense events result in lower energy

efficiencies than longer, lower intensity events. Given no other constraints (tectonic activity, geology, plants, climate, land use, etc.), this might imply that morphology organizes in time through erosion to facilitate faster and more intense runoff rates, for instance by forming rill- (Schroers, et al., 2022) and river networks. Both increase the power available for downstream sediment transport (Kleidon et al., 2013; Berkowitz and Zehe, 2020), while the local slope declines.

In the second part of the study, we tested whether similar behaviour can be found for extreme flood events in runoff and erosion

rates, observed in the Weiherbach catchment. We used a previously calibrated physical model (Catflow, cf. Zehe et al., 2001) to calculate relative dissipation, work and free energies of surface runoff and erosion for both extreme rainfall runoff events in 1994 and 1995. Surprisingly, we found a clear hierarchy of declining energy efficiencies with increasing morphological age for the three hillslope forms. Younger hillslopes, characterized by diffusive soil creep erosion receive largest free energy influxes from rainfall but dissipate in comparison to soil wash hillslope types less of this input, leaving relatively more free

energy available for erosion and sediment transport. While this was found for both events, we highlight that the hillslope system is generally energetically rather inefficient, although the well-known Carnot limit does not apply here.

We conclude that the energy efficiency of overland flow during events does indeed constrain erosional work and the degree of freedom for morphological changes. We conjecture that hillslope forms and overland dynamics coevolve, triggered by overshoot in power during intermittent rainfall runoff events, towards a decreasing energy efficiency in overland flow. This

means a faster depletion of energy gradients during events, and a stepwise downregulation of the available power to trigger further morphological developments, and this also implies the emergence of quasi-steady, metastable configurations, which optionally might maximize power in water and sediment fluxes, when averaged in space and time.





**Appendix A: Calculation of energy fluxes with hydraulic variables of overland flow**

Starting with a spatially distributed system along the flow path x, we separate the balance of potential energy flux into overland flow $J^{pe}_{f,net}$ plus rainfall $J^{pe}_{Peff}$ (cf. Schroers et al, 2022). The transient energy balance in watt per unit flow length can then be written as

$$D_f(x,t) = J^{pe}_{f,net}(x,t) + J^{pe}_{Peff}(x,t) + J^{ke}_{f,net}(x,t) - \frac{dE^{pe}_f(x,t)}{dt} - \frac{dE^{ke}_f(x,t)}{dt} \tag{A1}$$

Where each term can be expressed as a function of hydraulic flow variables h (water elevation above hillslope outlet in m), v (flow velocity in m s$^{-1}$) and Q (discharge in m$^3$ s$^{-1}$), hillslope segment width b in m, effective precipitation I in mm hr$^{-1}$, flow

density $\rho$ approximated as 1000 kg m$^{-3}$, and gravitational constant g approximated as 9.81 m s$^{-2}$:

$$J^{pe/ke}_{f,net}(x,t) = -div\left(J^{pe/ke}_f(x,t)\right) \qquad watt\ m^{-1} \tag{A2}$$

$$J^{pe}_f(x,t) = E^{pe}_{sp}(x,t)Q(x,t) = gh(x,t)\rho Q(x,t) \qquad watt \tag{A3}$$

$$J^{ke}_f(x,t) = E^{ke}_{sp}(x,t)Q(x,t) = \frac{v(x,t)^2}{2}\rho Q(x,t) \qquad watt \tag{A4}$$

$$J^{pe}_{Peff}(x,t) = \frac{\rho I(x,t)gh(x,t)b(x)}{3.6 \times 10^6} \qquad watt \tag{A5}$$

$$E^{pe}_f = \rho g \frac{Q(x,t)}{v(x,t)}h(x,t) \qquad joule\ m^{-1} \tag{A6}$$

$$E^{ke}_f = \frac{\rho}{2}Q(x,t)v(x,t) \qquad joule\ m^{-1} \tag{A7}$$

Leading to Eq. A8:

$$
\begin{aligned}
D_f(x,t) = \rho g &\left( -\frac{dQ(x,t)}{dx}h(x,t) - \frac{dh(x,t)}{dx}Q(x,t) + I(x,t)h(x,t)b(x)/(3.6 \times 10^6) \right) \\
&- \frac{1}{2}\rho\left( \frac{dQ(x,t)}{dx}v(x,t)^2 + 2v(x,t)\frac{dv(x,t)}{dx}Q(x,t) \right) \\
&- \rho g\left( \left( \frac{dQ(x,t)}{dt}\frac{1}{v(x,t)} + Q(x,t)\left( -\frac{\frac{dv(x,t)}{dt}}{v(x,t)^2} \right) \right)h(x,t) + \frac{Q(x,t)}{v(x,t)}\frac{dh(x,t)}{dt} \right) \\
&- \frac{1}{2}\rho\left( Q(x,t)\frac{dv(x,t)}{dt} + \frac{dQ(x,t)}{dt}v(x,t) \right)
\end{aligned}
\tag{A8}
$$



**Code availability**

The code of the Catflow model is available upon request.

**Data availability**

The used datasets (Gerlinger, 1999; Zehe, 1999) have been published by the KIT and is accessible through its library and is
available upon request.

**Supplement**

The supplemental code related to this article is available online at: https://github.com/shmulik1990/swe_cormack.git.

**Author contribution**

S. Schroers conceptualized, implemented the MacCormack scheme for the SWE, conducted the analysis and wrote the paper.
U. Scherer provided the original Catflow setups and commented on surface runoff dynamics. E. Zehe oversaw the study and
theory development as mentor.

**Competing interests**

At least one of the (co-)authors is a member of the editorial board of Hydrology and Earth System Sciences. The peer-review
process was guided by an independent editor, and the authors also have no other competing interests to declare.

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
