# Peer review of "Energy efficiency in transient surface runoff and sediment fluxes on hillslopes – a concept to quantify the effectiveness of extreme events"

_EGUsphere, 2022_

## Referee Comment (RC2)

The present paper is an extension of a previous work of the Authors regarding the analysis of the runoff process at the hillslope scale in a energy perspective. The extension concerns the inclusion of the transient regime, with respect to the previous steady stat analysis. This is crucial since intermittent rain events provide potential energy to the open hillslope system which can be either dissipated or stored. These energy dynamics are linked to characteristic geomorphology aspects of hillslopes in search of a connection between energy balance and the erosion process.

The paper is really interesting and presents an innovative perspective to the problem. I enjoyed reviewing it. At the same time, I think it could be clearer and more accessible after few comments listed below are addressed.

**Comment 1**

From Eq. (2) it is my understanding that $D_{HS}(t)$ represents the amount of kinetic energy of the surface runoff that is dissipated. This dissipation can be of different natures, e.g., viscous and turbulent dissipations or erosion of the soil.

Then in Eq 7b
$$\widehat{D}_{HS} = 1 - \hat{J}_{HS} - \widehat{E}_{HS} = \widehat{A}_{HS} - \widehat{E}_{HS}$$
where $\widehat{A}_{HS}$ is the available cumulated amount of energy (i.e., potential energy from the rain event minus the kinetic energy that has left the system at the closer section) and $\widehat{E}_{HS}$ is the stored potential + kinetic energy of the surface runoff (i.e., the sum of the area below the curves in Fig. 5a and 5b for a given scenario).

At this point, my understanding is that a high value of $\widehat{E}_{HS}$ is indicative of a surface runoff with a high energy content, e.g., high velocities with a high kinetic energy component or high water levels which favor the potential energy component. That it is, high $\widehat{E}_{HS}$ means high energy in the surface water, i.e., less energy is available to be dissipated. But at lines 422-425 it is stated that "Interestingly our results show that the latter (SW) results in less energy efficiency of surface runoff, or differently stated a larger fraction of the provided free energy by rainfall is dissipated than for SC hillslope types (cf. Fig. 6b). This means that there is relatively more energy available for work on the surface of SC profiles (be it in the form of detachment or transport of sediment particles)." This is to say that small values of kinetic dissipation $\widehat{D}_{HS}$ favors the erosion of soil, but the above description was pointing towards the idea that higher dissipation of kinetic energy would favor dissipative processes associated with the flow (i.e., kinetic energy) of the surface water either that being turbulent losses or soil erosion. In the proposed framework there is not a distinction on the type of kinetic energy dissipation. It is true that smaller values of $\widehat{D}_{HS}$ means more relative energy in the flowing water $\widehat{E}_{HS}$ (as a sum of potential and kinetic components), but that it is energy possess by the of the surface water while $\widehat{D}_{HS}$ embeds what is lost/dissipated without specifying in which form is lost, e.g., turbulent dissipation? sediment transport?

Shouldn't be a differentiation on the components of dissipated kinetic energy whit a focus on the part available to sediment erosion/transport?

**Comment 2**

Lines 422-425: "Interestingly our results show that the latter (SW) results in less energy efficiency of surface runoff, or differently stated a larger fraction of the provided free energy by rainfall is dissipated than for SC hillslope types (cf. Fig. 6b). This means that there is relatively more energy available for work on the surface of SC profiles (be it in the form of detachment or transport of sediment particles)." Lines 609-611: "In the next section we do not only confirm this general erosion pattern but also show that highest erosion rates coincide with highest relative dissipation rates and therefore maximum work which overland flow performed on the sediments"

Aren't these two statements in contradiction? What am I missing?

**Comment 3**

In the numerical experiment the profile of the hillslope is given and the hydrodynamic part is solved. If the Author are testing (Lines 260-261) " … we test our hypothesis that the evolution of landscape forms is directly linked to energy efficiency of transient overland flow events" should not the hillslope profile co-evolve with the transient overland flow?

**Comment 4**

How adding the groundwater compartment would affect the framework? Would that impact the assumption that (Lines 221-222) "In this case we regard the potential energy which enters the system much larger than the potential energy which leaves the system and therefore also $J^{pe}_{HS,out}$ $(t)$ (watt) to be negligible" since much of the water could leave the system as infiltrating groundwater?

**Comment 5**

After introducing Eq. (5) there is an interesting discussion in Lines 229-243. However, it could help to introduce the definition of free energy and power in terms of the quantities in Eq. (5) to help the reader. For example, it is hard to grasp the meaning of Lines 236-237 "If a system receives a certain amount of energy influx, it is therefore clear that optimization must happen through adjustment of the internal spatial structure which determines temporal derivatives of free energy conversion rates" without a definition of free energy and free energy conversion rates clearly stated.

**Minor comments**

Lines 276-277: "In its simplest form sediment transport capacity C is at least dependant on accumulated discharge and local gradient $C=Q^m \times S^n$" please specify what is Q and S. Moreover, local gradient of what?

Lines 212-213: "Eq. 1 to 4 are a simplification of surface runoff, as we do not consider other types of energy than potential and kinetic energy of water" maybe better after Eq. (4).

Line 335: "where y is the averaged variable in time" should not be f(y)?

Caption of Figure 3: "in space is integral over time (blue)" should not be (yellow)? The same in the text at line 343.

The text appears to be sometime a bit convoluted. For example, Lines 47-48 "At the hillslope scale, one can depending on the morphological age of the system observe typical hillslope forms." Might be better formulated as "At the hillslope scale, depending on the morphological age of the system, one can observe typical hillslope forms". Another example, Lines 93 "while the vast majority has dissipated at the downstream/downslope outlet." Should not be Is dissipated?. Lines 124-125: "Moreover, maximum power in the combined sediment-water flux does in steady state correspond to maximum entropy production." The sentence would flow better as "Moreover, maximum power in the combined sediment-water flux corresponds to maximum entropy production during steady state condition". Line 155: "but also relates to larger friction coefficients which in turn limit overall energy efficiency" limit THE overall energy efficiency. Lines 303-304: "which has been set to equal 1.0 for a uniform velocity distribution." Should not be 'has been set to 1.0' or 'has been set equal to 1.0'?

Lines 53-55: "However, despite of these obstacles, there has been continuous research to discover the seemingly hidden physical laws governing and constraining the co-development of form and functioning of the Earth's hydrologic systems" It is my understanding that 'these obstacles' are the semi-empirical relationships and the use of tunning parameters that are mentioned earlier in the text. Yet, when the latter are introduced the fact that these are problematic might not be clear to all the readers, try to emphasize why is that so.

---

## Author Comment (AC1)

Efficiency of flow

The Manning coefficient, which currently is for most applications the only way to pinpoint efficiency of free energy conversion that scales dissipation. Any parameter which characterizes roughness is ultimately related to the conversion process of free energy to heat, describing the capacity of the system to create flow from a gradient of free energy. In fact, expressing this in steady state as the flux of kinetic energy $J_f^{ke}$ over stream power $P$:

$$E_Q = \frac{J_f^{ke}}{P} = \frac{\frac{1}{2}\rho Q v^2}{\rho g S Q} = \frac{v^2}{2gS}$$

with a formula that links average flow velocity and driving gradient such as the Manning equation:

$$S = \left(\frac{vn}{R^{\frac{2}{3}}}\right)^2$$

$E_Q$ becomes:

$$E_Q = \frac{R^{\frac{4}{9}}}{2gn^2}$$

The efficiency of a system to convert a gradient of free energy into kinetic energy is therefore expressed as a function of geometry (hydraulic radius) and roughness.

---

## Author Response (AR1)

Response review #1:

The reviewer correctly points out that the focus of our preceding study and the study at hand lies on dissipation of free energy. This already partly explains the reviewer's comment regarding a definition of stream power, which from our point of view is the magnitude of potential energy (of water) that is being converted into other forms of energy per unit time. It is correct that we have made a mistake in Fig. 1, where we neglected the dissipation term, we have corrected this figure.

Eq. 2 outlines, that dissipation is converted energy (stream power) minus the amount of energy which remains in the form of kinetic energy. Stream power alone therefore informs about the magnitudes of energy that is being converted, together with kinetic energy (and other remaining free energy which we have not considered in this study, e.g., turbulent kinetic energy) one can assess the efficiency of the conversion of a free energy gradient to flow, the movement of water.

This points to the next comment from the reviewer regarding the Manning coefficient, which currently is for most applications the only way to pinpoint this efficiency that scales dissipation. Any parameter which characterizes roughness is ultimately related to the conversion process of free energy to heat, describing the capacity of the system to create flow from a gradient of free energy. In fact, expressing this in steady state as the flux of kinetic energy over stream power with a formula that links average flow velocity and driving gradient such as the Manning equation we can derive a formula which describes the efficiency of a system to convert a gradient of free energy into kinetic energy as a function of geometry (hydraulic radius) and roughness.

This leads again to our argumentation that to understand the evolution of the dynamics of flow, e.g., in a geomorphological context for erosion of hillslopes, we need to understand the evolution of efficiencies. We argue that the underlying driving process for evolution of the structure of a system does not depend on the physical parameter of roughness but on dissipation itself.

The comment from the author regarding negative dissipation of a single hillslope in Fig. 11 is correct. We traced this to the implementation of very shallow flows in the numerical scheme, an error we corrected by allowing smaller minimum water depths for movement of water.

Regarding the rest of the comments from the reviewer we thank her/him for the thorough inspection and we have gladly incorporated the suggestions regarding readability and clarity.

We thank the reviewer for his effort and comments.

All minor comments have been addressed in the revised manuscript.

Response review #2:

We would like to answer comments 1 and 2 in one paragraph, as we believe both are related and belong together:

The reviewer correctly highlights that the calculated dissipation term does not differentiate between the type of dissipation, be it the creation of turbulence, the lift or the transport of sediment particles. First, we would like to point to our previous publication (Schroers et al., 2022) and in particular to the discussion we had with Keith Beven regarding the same issue (https://doi.org/10.5194/hess-2021-479-RC1). Among other things we presented in https://doi.org/10.5194/hess-2021-479-AC2 an extension of our theoretical framework to distinguish the energy, which is spent on erosion, but this usually goes beyond what is possible to reliably represent with field data. There are however several studies (e.g. Emmett, 1970) which estimated the type of flow regime (laminar or turbulent) on which we have elaborated in our previous study. Maybe the most interesting result is that the build-up of free energy seems to be related to laminar flow and the decrease to turbulence. As turbulence is further related to higher erosion

rates, we hypothesized that the occurrence of erosional structures such as rills or gullies can be pinpointed by the free energy content of surface runoff.

On a larger scale the hillslope itself is shaped into a certain form (SC or SW), typically by intermittent surface runoff events. This led us to the idea to analyze transient events in the study at hand. We therefore defined energy efficiency of a hillslope in line with energy efficiency of a mechanical machine, the output of free energy divided by the input of free energy. A more efficient surface runoff event is therefore one which allows a larger fraction of the input energy to be conserved in the energy output. Our results show that for transient events higher efficiency typically relates to SC hillslope types and lower efficiency to SW hillslope types. In a second step we argue that a higher efficiency is downregulated through erosion to smaller efficiency (the typical evolution of hillslopes from SC to SW forms). Section 4.4 shows that this reasoning does indeed apply to the hillslopes and surface runoff events in the Weiherbach catchment. Higher relative dissipation (SW forms, less efficiency) relates to less erosion and smaller relative dissipation relates to more erosion (SC forms, higher efficiency). It is therefore correct that we made a wording mistake in line 609-611, we have corrected this sentence.

Comment 3:

We agree with the reviewer, ideally the hillslope should coevolve with the transient event. However, it was shown elsewhere (e.g. Kirkby, 1971) that hillslopes generally evolve from SC to SW profiles. In this perspective our tests consider only the beginning and the end of this evolution and we subsequently present the different energy fluxes in the presented thermodynamic framework. Our approach is therefore a simplification to highlight the differences between the start and the end point and provide a thermodynamic explanation to the direction of such hillslope evolution.

Comment 4:

We thank the reviewer for pointing out that infiltration could potentially have a large effect on the presented framework. In theory high infiltration rates would decrease the free energy of surface runoff, but at the same time it would increase the free energy of subsurface water. In this study we put our focus on surface runoff events, but the free energy content of subsurface water could certainly correlate with observed hillslope forms. As we also point out in line 546-549, Zehe et al. (2013) found for the same events in the Weiherbach catchment energy conversion rates of almost the same scale for subsurface runoff as we found in surface runoff. This highlights that surface and subsurface runoff of extreme events are likely to be co-organized. Such an analysis is however beyond the scope of this work.

All minor comments have been addressed in the revised manuscript.